# Exploring the evolution and function of Canoe's intrinsically disordered region in linking cell-cell junctions to the cytoskeleton during embryonic morphogenesis

Noah J. Gurley[1☯], Rachel A. Szymanski[1☯], Robert H. Dowen[1,2,3,4], T. Amber Butcher[1], Noboru Ishiyama[5], Mark Peifer[1,2]*

1 Department of Biology, University of North Carolina at Chapel Hill, Chapel Hill, NC, United States of America, 2 Curriculum in Genetics and Molecular Biology, University of North Carolina at Chapel Hill, Chapel Hill, NC, United States of America, 3 Integrative Program for Biological and Genome Sciences, University of North Carolina at Chapel Hill, Chapel Hill, NC, United States of America, 4 Department of Cell Biology and Physiology, University of North Carolina at Chapel Hill, Chapel Hill, NC, United States of America, 5 Launchpad Therapeutics, Inc., Cambridge, MA, United States of America

☯ These authors contributed equally to this work.
* peifer@unc.edu

**Data Availability Statement:** All relevant data are within the paper, Figures and Tables.

## Abstract

One central question for cell and developmental biologists is defining how epithelial cells can change shape and move during embryonic development without tearing tissues apart. This requires robust yet dynamic connections of cells to one another, via the cell-cell adherens junction, and of junctions to the actin and myosin cytoskeleton, which generates force. The last decade revealed that these connections involve a multivalent network of proteins, rather than a simple linear pathway. We focus on *Drosophila* Canoe, homolog of mammalian Afadin, as a model for defining the underlying mechanisms. Canoe and Afadin are complex, multidomain proteins that share multiple domains with defined and undefined binding partners. Both also share a long carboxy-terminal intrinsically disordered region (IDR), whose function is less well defined. IDRs are found in many proteins assembled into large multiprotein complexes. We have combined bioinformatic analysis and the use of a series of *canoe* mutants with early stop codons to explore the evolution and function of the IDR. Our bioinformatic analysis reveals that the IDRs of Canoe and Afadin differ dramatically in sequence and sequence properties. When we looked over shorter evolutionary time scales, we identified multiple conserved motifs. Some of these are predicted by AlphaFold to be alpha-helical, and two correspond to known protein interaction sites for alpha-catenin and F-actin. We next identified the lesions in a series of eighteen *canoe* mutants, which have early stop codons across the entire protein coding sequence. Analysis of their phenotypes are consistent with the idea that the IDR, including the conserved motifs in the IDR, are critical for protein function. These data provide the foundation for further analysis of IDR function.

**Funding:** This work was supported by National Institutes of Health R35 GM118096 to M.P https://www.nih.gov/ The funders had no role in study design, data collection and analysis, decision to publish, or preparation of the manuscript. Work in the Dowen lab is supported by National Institutes of Health R35 GM137985 to R.H.D. https://www.nih.gov/ The funders had no role in study design, data collection and analysis, decision to publish, or preparation of the manuscript.

**Competing interests:** The authors have declared that no competing interests exist.

## Introduction

The most common tissue organization in animal bodies is the epithelium, which, in its simplest form, is a single-cell thick sheet of cells with defined apical and basal surfaces. To build epithelia, cells evolved the ability to adhere to one another and to the extracellular matrix that they or their neighbors secrete. Few epithelia are static—even in adults they are constantly remodeled, with new cells replacing those lost to injury or apoptosis, and often with cells moving with respect to their neighbors. This is even more dramatic during embryonic development, when epithelia bend into tubes, elongate tissues by cell rearrangement, or collectively migrate [1]. These processes all require force production, often generated by myosin motor proteins walking along actin filaments. However, for the force generated to drive cell shape change or movement, the cytoskeleton must be linked to the plasma membrane. These linkages occur at cell-cell and cell-matrix junctions.

In cell-cell adherens junctions (AJs), homophilic interactions between cadherin extracellular domains join neighboring cells, and cadherin cytoplasmic domains organize a multiprotein complex that mediates interaction with the actin cytoskeleton. Beta- and alpha-catenin are essential for adhesion itself, and alpha-catenin mediates mechanosensitive linkage to actin [2]. Other proteins strengthen this connection—among these are *Drosophila* Canoe (Cno) and its vertebrate homolog Afadin. In both *Drosophila* and mice, Cno/Afadin facilitates completion of morphogenetic movements of gastrulation, and *Drosophila* Cno plays roles in diverse events of morphogenesis. Cno strengthens AJs under elevated force, such that in its absence the cytoskeleton detaches from AJs and gaps appear at apical junctions [3–6]. Cno is a complex, multidomain protein with many predicted binding partners in the network of proteins linking AJs and the cytoskeleton. To further understand Cno's mechanism of action, we are exploring how different parts of the protein contribute to its diverse functions.

Much of the initial analysis of protein function focused on defining the roles of structured domains. These fold into stable structures that mediate protein interactions and most enzymatic functions. The need to conserve protein folding constrains divergence of these domains, and orthologs can often be recognized over long evolutionary distances, such as that between insects and vertebrates. Cno and Afadin share a series of such folded domains (Fig 1A). Each has two N-terminal Ras association (RA) domains that bind the small GTPase Rap1 when it is

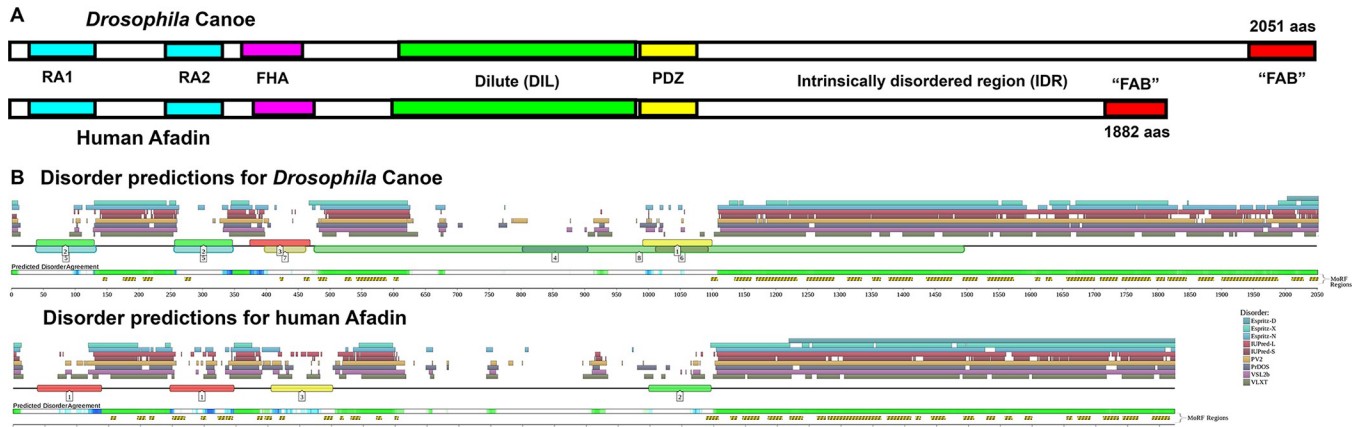

**Fig 1.** *Drosophila* **Canoe and human Afadin shared conserved folded domains and a long C-terminal intrinsically disordered region.** A. Diagram to scale of *Drosophila* Canoe and Human Afadin. B. Output of D2P2, the database of disordered protein predictions. Predicted folded domains are numbered boxes along the line. Above the line are disorder predictions from different programs, and below the line are regions of Predicted Disorder Agreement and sequences defined by the software as molecular recognition features (MoRFs).

in its active GTP-bound form; this binding "activates" Cno, though the mechanisms involved remain mysterious. Following this are two domains defined only by their similarity to known domains of other proteins: a Forkhead associated (FHA) domain, known in other proteins to bind phosphothreonine-containing peptides, and a Dilute (DIL) domain, found in unconventional myosin V family members. The DIL domain is the predicted binding site for the protein ADIP [7], but the functions of both domains in Cno remain untested. More C-terminal is a PSD-95/Discs Large/ZO-1 (PDZ) domain that binds the transmembrane AJ proteins E-cadherin and nectins. Mutational analysis of *cno* revealed that the RA domains play a critical role in Cno function, with their removal nearly eliminating protein function. However, we were surprised to find that the PDZ domain, despite conservation of its sequence, structure, and binding partners, is largely dispensable for Cno function [8].

As the universe of known and predicted protein domains expanded, it became clear that many proteins also contain, and are sometimes entirely composed of, sequences that are intrinsically unstructured, now called intrinsically disordered regions (IDRs). This definition covers a broad spectrum of protein regions without a strong unifying theme. IDRs often have reduced sequence complexity, but vary widely in amino acid sequence composition and charge. They also vary in their level of sequence conservation, with some diverging relatively rapidly over short evolutionary distances. Perhaps as a result, diverged IDRs located at similar positions in homologous proteins vary in their ability to rescue function when interchanged. IDRs are implicated in conferring many protein properties, including the ability to phase separate and form biomolecular condensates [9–11].

In some IDRs, while protein sequence is not maintained over evolutionary time, conservation of other features such as isoelectric point, hydrophobicity, net charge, or acidic residue content is important for function, potentially by helping regulate protein localization or interactions [12, 13]. Many IDRs have embedded within them short sequences (10–70 amino acids) that are known or suspected protein binding sites, referred to as Short Linear motifs (SLiMs; [14]) or Molecular Recognition Features (MoRFs; [15]). Some of these sequences can assume secondary structures in isolation. Others undergo a disorder-to-order transition upon binding their partners. Scientists use evolutionary conservation as a tool to identify these motifs [14]. In some proteins these peptide motifs are well-conserved over long evolutionary distances. For example, the tumor suppressor protein APC has multiple short peptide motifs whose sequences and functions are broadly conserved across different animal phyla—these are binding sites for its protein binding partners Axin (the SAMP motifs), beta-catenin (the 15 and 20 amino acid repeats), or alpha-catenin (the CID or R2/B motif) [16]. In other proteins, motifs are conserved at the level of order or phyla but diverge over longer evolutionary distances. For example, in the non-receptor tyrosine kinase Abelson four motifs in the IDR are conserved in insects, but only one of these is also conserved between insects and vertebrates, and no single motif underlies the function of the full-length IDR [17, 18]. Other IDRs contain few or no apparent conserved motifs.

Cno and its mammalian homolog Afadin each terminate in long regions presumed to be IDRs (Fig 1A). Their functions remain incompletely defined. Both Cno and Afadin can interact directly with actin filaments via regions that include part of the IDR [4, 19], but the precise region that binds F-actin is controversial [20–22]. Afadin's IDR includes a motif that can bind alpha-catenin, and deleting this motif alters the ability of Afadin to bundle actin at the AJ [20]. ZO-1 interacts with proline-rich regions of the IDR using its SH3 domain [23]. The IDR also contains a peptide that binds the tetratricopeptide repeat (TPR) domain of Lgn, and this peptide plays a role in spindle orientation in HeLa cells [21]. Intriguingly, although there is a conserved physical interaction between *Drosophila* Cno and Lgn's fly homolog, the peptide that co-crystalizes with Afadin is not conserved in *Drosophila*. Further, none of these regions has

been assessed for function during embryonic development or tissue homeostasis. The IDR sequences that are best conserved between Cno and Afadin are the C-terminal sequences referred to as the F-actin binding (FAB) region, based on the mapping of this property by the Takai group [22]. We were surprised to learn that the FAB region is not essential for Cno function, but it supports Cno's role in strengthening AJs under tension [8]. Thus, the role of the IDR overall, and the conservation and function of binding sites within it have not been explored in detail. We sought to learn more about the potential roles of the Cno and Afadin IDRs, combining bioinformatic analysis and a series of *cno* mutants that are predicted to truncate Cno at different points within the IDR or earlier.

## Results

### The IDRs of *Drosophila* Cno and human Afadin have diverged in length, sequence, amino acid composition, and charge

*Drosophila melanogaster* Cno and human Afadin share identical domain architectures (Fig 1A), with two N-terminal RA domains, followed by FHA, DIL and PDZ domains (below, if not otherwise stated, *Drosophila* refers to *Drosophila melanogaster*). Between *Drosophila* and humans, these domains are variably conserved in sequence: our analysis revealed that while the RA1 and the PDZ domains both have > 70% amino acid identity, the RA2, FHA, and DIL domains are less strongly conserved (52%, 44%, and 48% amino acid identity, respectively; Fig 2). Both Cno and Afadin can bind F-actin [4, 19], but the precise region that binds actin is controversial, with different groups attributing actin-binding to different regions of the IDR [20–22]. The region we have referred to as the F-actin binding region (FAB), which is at the C-terminus of Canoe/Afadin, includes 115 amino acids that are reasonably well-conserved among insects (e.g. 61% identical between *Drosophila* and the beetle *Tribolium*; Fig 2) or among vertebrates (56% identical between human and zebrafish; Fig 2), but the FAB region is considerably less well-conserved between *Drosophila* and humans (<35% identical, even when ignoring multiple insertion/deletions (indels; [8]).

In between the PDZ domain and the FAB both *Drosophila* Cno and human Afadin have a long region that we previously referred to as the intrinsically disordered region (IDR) because of its lack of predicted protein domains and its low sequence complexity. To confirm that these regions fit the established criteria for IDRs, we used D2P2, the database of disordered protein predictions, which provides a community resource of precalculated predictions for proteins from multiple animal models, comparing multiple prediction methods [24]. The regions following the PDZ domains of *Drosophila* Cno and human Afadin, including the region we referred to as the FAB, are both confidently predicted to be disordered by multiple algorithms (Fig 1B). There are also shorter regions of predicted disorder between the RA domains, between the RA2 and FHA domains, and between the FHA and DIL domains (Fig 1B).

Despite their shared domain architecture and predicted C-terminal IDRs, the IDRs of the *Drosophila* and human orthologs are strikingly different. The Cno IDR is substantially longer–837 versus 705 amino acids (Fig 1A and 1B)–and they have very little sequence similarity. Our search of the human proteome using the NCBI blast algorithm and the *Drosophila* IDR as a query yielded "no significant similarity" outside the FAB region, and the reciprocal search of the *Drosophila* proteome using the human IDR yielded the same result. When we attempted a pairwise blast alignment of these *Drosophila* and human IDRs, the algorithm only aligned 140 amino acids, and in this region the amino acid identity was only 29%.

Relatively low conservation of amino acid sequence is a feature of many IDRs, which are often rich in low complexity sequence repeats. However, sequence composition and charge are

| Comparison | Estimated divergence time | RA1 112 aa | RA2 91 aa | FHA 101 aa | DIL Fly 375 aa | DIL Human 387 aa | PDZ 94 aa | FAB Fly 115 aa | FAB Human 97 aa | IDR Fly 837 aa | IDR Human 705 aa |
|---|---|---|---|---|---|---|---|---|---|---|---|
| **Conservation of folded domains** | | | | | | | | **Conservation of the FAB and IDR** | | | |
| **D. mel vs human** | | | | | | | | | | | |
| D. mel vs human | | 74.10% | 53.8% +3 & 7 aa indels | 41.58% +6 aa indel | 47.6% +3 & 8 aa indels | | 71.27% | 34.78% +3, 5, & 7 aa indels | 22.68% +3, 5, & 7 aa indels | "no significant similarity" | "no significant similarity" |
| **Comparisons Among Insects** | | | | | | | | | | | |
| D. mel vs D simulans | 3 million years | 100% | 98.90% | 100% | 100% | | 100% | 99.10% | NA | 96.42% | NA |
| D. mel vs D yakuba | 5–6 million years | 100% | 97.80% | 100% | 98.90% | | 98.90% | 95.60% | NA | 86.26% | NA |
| D. mel vs D psuedoobscura | 30 million years | 100% | 100% | 99% | 99.20% | | 98.90% | 98.30% | NA | 81.72% | NA |
| D. mel vs D virilis | 50 million years | 100% | 98.90% | 100% | 97.60% | | 97.90% | 96.52% | NA | 74.55% | NA |
| D. mel vs Musca | >70 million years | 98.20% | 97.80% | 94.10% | 96.30% | | 93.60% | 85.20% | NA | 50.29% | NA |
| D. mel vs Bradysia | 70–200 million years | 86.60% | 94.50% | 88.10% | 81.60% | | 93.60% | 73.90% | NA | 37.40% | NA |
| D mel vs Anopheles | 150–200 million years | 87.50% | 94.5% +6 aa indel | 84.10% | 85.01% | | 89.36% | 79.09% +3 & 4 aa indels | NA | blast only pairs 102 aas near PDZ | NA |
| D. mel vs Heliconius | 290 million years | 77.67% | 72.50% | 48.5% +3 aa indel | 59.46% +3 aa indel | | 80.85% | 66.09% +3, 4, & 5 aa indels | NA | "no significant similarity" | NA |
| D. mel vs Tribolium | 330 million years | 83.90% | 79.12% +4 aa indel | 55.44% +6 aa indel | 59.73% +6 aa indel | | 80.85% | 60.87% + 8 aa indel | NA | "no significant similarity" | NA |
| Tribolium vs Heliconius | | 80.36% | 70.33% +3 aa indel | 52.48% +3 aa indel | 57.33% +4 & 8 aa indels | | 78.72% | 66.09% +3 aa indel | NA | 45.80% in region Blast aligned 1059/1176 aas | NA |
| **Comparisons Among Vertebrates** | | | | | | | | | | | |
| Human vs mouse | 80 million years | 100% | 96.70% | 95.04% | | 98.19% | 100% | NA | 78.35% | NA | 85.92% |
| Human vs zebra finch | 325 million years | 97.32% | 94.50% | 97.03% | | 94.31% +7 aa indel | 98.93% | NA | 75.26% +3 aa indel | NA | 83.83% |
| Human vs Xenopus | 350 million years | 96.42% | 83.50% | 84.58% | | 91.47% | 93.61% | NA | 62.89% | NA | 71.77% |
| Human vs zebrafish | 450 million years | 96.42% | 80.22% +6 aa indel | 75.25% +14 aa indel | | 89.14% | 94.68% | NA | 55.67% | NA | 69.79% |
| Human vs lamprey | | 84.82% | 54.45% | 70.30% | | 66.90% | 86.17% | NA | 46.39% +3 aa indel | NA | 40.57% |
| Human vs Amphioxus | | 75.00% | 57.14% | 55.54% | | 59.6% +6 & 39 aa indels | 77.66% | NA | 36.08% +3 aa indel | NA | "no significant similarity" |

Green = 85-100% Identical    Blue=70-84% identical
Yellow=55-69% identical    Red=<55% identical

**Fig 2. Amino acid conservation of the folded domains and IDRs of Cno and Afadin.** For each folded domain and for the IDR and FAB we calculated amino acid identity. Comparisons were made relative to Drosophila melanogaster Cno (relative to other insects—all except Tribolium and Heliconius are in the Order Diptera) or human Afadin (relative to other vertebrates and one non-vertebrate chordate, Amphioxus). Color coding indicates ranges of sequence identity. Green = 85–100% identical. Blue = 70–84% identical. Yellow = 55–69% identical. Red = <55% identical. Small insertion/deletion polymorphisms (indels) are noted. "No significant similarity" means the program Blast found no match.

sometimes conserved even when sequence identity diverges—similarities in these properties, even in the absence of protein sequence identity, can sometimes maintain protein localization and function [12, 13]. We thus examined whether the *Drosophila* and human IDRs share these features. Strikingly, we found that the two IDRs are quite different in both properties. The *Drosophila* IDR is most enriched in glutamine, serine, and asparagine (16.7%, 10.5%, and 9%, respectively) while the human IDR is most enriched in proline, glutamic acid, and arginine (11.5%, 10.9%, and 9.8%, respectively). They also differ strikingly in the percentage of charged amino acids: 13.5% in *Drosophila* and 32.3% in humans.

*Drosophila* Cno is alternatively spliced (flybase.org) and some of the alternative splice isoforms differ in the IDR. In our analysis below, we used CnoRE, which encodes the longest isoform, as flies with their *cno* gene engineered to encode only this isoform are viable, fertile and wildtype in phenotype [8]. Other isoforms differ. CnoRH lacks exon 16, removing 84 amino acids near the beginning of the IDR. However, two other isoforms, CnoRD and CnoRI, differ more dramatically. They insert a long alternative exon after exon 15, thus replacing almost the entire IDR and FAB region with an alternative 718 amino acid sequence. RNA-Seq data available from flybase.org supports the idea that these alternative splice variants are expressed,

though not at the same level as CnoRE or other "canonical" IDR isoforms. The alternative IDR bears no detectable similarity to the "canonical" Cno IDR, as assessed by Blast alignment, and also has no detectable similarity to the human Afadin IDR. It differs in sequence composition from both—it is strongly enriched in serine (14.2%), followed distantly by arginine and alanine (7.7% and 7.1%, respectively). This alternative IDR is even less well-conserved than the canonical one. In the closely related *Drosophila pseudoobscura*, the canonical IDR remains 82% identical while the alternative IDR is only 64% identical. Human Afadin is also predicted to encode multiple splice isoforms, some of which have altered IDRs—for example, one which has begun to be functionally characterized is s-Afadin, which lacks the C-terminal 174 amino acids (aas), including the FAB region [19, 22]. Thus, while both *Drosophila* Cno and human Afadin terminate in long IDRs that can be alternatively spliced, their sequences and sequence properties are dramatically divergent.

### Comparisons of IDRs reveals strong sequence conservation over shorter evolutionary distances that drops off after longer lineage divergence, while folded domains remain well conserved

How did these striking differences arise, with selection maintaining the presence of a long IDR but not its sequence or even sequence composition? To address this, we compared divergence in the folded protein domains of Cno/Afadin with that of the IDR over shorter evolutionary intervals than that between fly and human. To do so, we used the Multiple Sequence Alignment program Clustal Omega [25, 26] from EMBL/EBI. We first compared Cno orthologs in insects.

Different *Drosophila* species are estimated to have diverged over time frames ranging from 3 million to 50 million years ago [27]. We found that all the folded protein domains are highly conserved over this time frame (≥98% identity; Fig 2). The FAB region is also strongly conserved (>96% identity; Fig 2). Cno's folded domains also remained conserved over longer evolutionary time scales, with ≥82% identity in more distantly related flies (other Dipterans like *Musca*, *Bradysia*, or *Anopheles*; Figs 2 and 3). In contrast, the IDRs diverged more rapidly. In comparisons of *Drosophila melanogaster* and *Drosophila yakuba*, estimated to have diverged 5–6 million years ago, IDR sequence identity is already reduced to 86% and when comparing *Drosophila melanogaster* and *Drosophila virilis*, estimated to have diverged 50 million years ago, IDR percent identity is only 75% (Fig 2). When one compares *Drosophila melanogaster* to more distantly related Dipterans [28], the divergence of the IDR is even more dramatic. Relative to the housefly *Musca domestica* (>70 million years estimated divergence), the folded protein domains remain highly conserved (all ≥94% identical), but IDR conservation is reduced to 50% identity (Fig 2). In the even more distantly related fungus gnat *Bradysia*, folded domains retain 82–94% identity, while the IDR retains only minimal conservation (37% identity; Fig 2). Finally, when we compared *Drosophila melanogaster* Cno to its orthologs in other insect orders [29]—the Lepidopteran *Heliconius* (estimated divergence 290 million years) or the Coleopteran *Tribolium* (estimated divergence 330 million years)—BLAST searches detected no significant similarity between the IDRs, though the FAB regions retains sequence similarity (Fig 2). At this evolutionary distance, some folded domains also began to diverge. The FHA and DIL domains are least conserved (55% and 60% sequence identity between *Drosophila* and *Tribolium*, with short insertion/deletion differences (indels) appearing) while RA1 is the most conserved (84% identical; Fig 2), and the PDZ and RA2 domains are also well conserved (81% and 79% identical, respectively; Fig 2). Thus, while the IDR is conserved in the *Drosophila* genus it is much less well-conserved in insects over longer evolutionary time scales.

We next examined the divergence of the Afadin IDR in the vertebrate lineage. Once again, we found that the IDRs diverged more rapidly than the folded domains, but the rate of overall

## Conservation of Cno's folded domains in Dipteran Insects

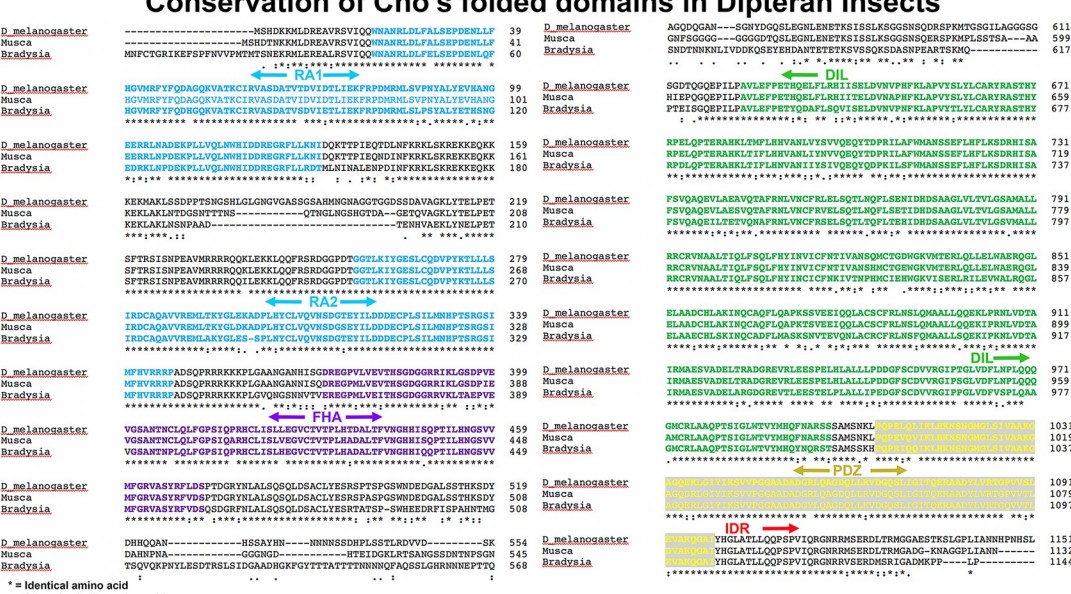

**Fig 3. Clustal sequence alignments of the N-terminal folded domains of Cno and its orthologs from selected Dipteran insects reveals strong conservation.** Asterisks indicate identical amino acids, colons and periods conservation between groups of strongly similar or weakly similar properties, respectively, and dashes gaps in the alignment. Predicted folded domains are indicated and highlighted in different colors.

protein divergence was substantially slower. Relative to insects, the folded protein domains were even more well-conserved across the vertebrate lineage (Figs 2 and 4): comparing human and zebrafish (450 million years estimated divergence; [30]), the least well-conserved was the FHA domain (75%) and all of the other folded protein domains were 80–96% identical (Figs 2 and 4). This is consistent with the known slower rate of protein evolution in the vertebrate clade compared to insects [31]. We found that the IDR diverged faster than the folded domains in the vertebrate lineage. After 80 million years of divergence, the mouse and human IDRs remained 86% identical (Fig 2). Human and bird IDRs (zebra finch; 325 million years estimated divergence) were 84% identical, human and amphibian IDRs (*Xenopus*; 350 million years estimated divergence) were 72% identical, and even human and bony fish IDRs (zebrafish, 450 million years estimated divergence) still shared 70% identity (Fig 2). There was even detectable similarity between the IDRs of human and the lamprey, a jawless fish (41% identity; Fig 2). However, we detected no significant matches using blast between the human IDR and that of the non-vertebrate chordate *Amphioxus*, although the folded protein domains retained 56–78% identity (Fig 2). Thus, in the vertebrate lineage, the IDR diverges more rapidly than the folded protein domains, but the IDR remains substantially more well-conserved throughout the clade than was observed in insects. This conservation does not extend into other chordates.

### Sequence comparison reveals conserved motifs in both insect and vertebrate IDRs, a subset of which are predicted to be alpha-helical by AlphaFold

Many IDRs contain embedded motifs that can serve as binding sites for other proteins. In some cases, these can be identified by examining conservation across phylogenetic groups—

# Conservation of Afadin's folded domains in Vertebrates

**Fig 4. Clustal sequence alignments of the N-terminal folded domains of human Afadin and its orthologs from selected vertebrates reveals strong conservation.** Asterisks indicate identical amino acids, colons and periods conservation between groups of strongly similar or weakly similar properties, respectively, and dashes gaps in the alignment. Predicted folded domains are indicated and highlighted in different colors.

for example, in Abelson kinase we identified four motifs conserved among insects, two of which had features suggesting they were binding sites for known protein partners, and one of which was also conserved in mammalian Abelson [17]. We thus examined whether there were conserved motifs in either insect or vertebrate Cno/Afadin IDRs.

In the vertebrate IDRs (70% identical overall from humans to zebrafish) we identified multiple conserved motifs (Fig 5; highlighted in yellow or green); these were defined as motifs of 15 aas or more that were 63–83% identical in sequence). There was very strong conservation of the region of the IDR immediately after the PDZ domain (74% identity in the first 151 aas), and there were eight additional motifs that fit our criteria (Fig 5; highlighted in yellow or green). We next examined insect IDRs. Due to the relatively rapid sequence divergence of the IDR in the insect lineage, we focused on comparisons among Dipterans, including different *Drosophila* species as well as the housefly *Musca domestica* and the fungus gnat *Bradysia coprophila*, among which overall IDR conservation was only 37%. We observed strong conservation

# Conservation of protein motifs and predicted alpha-helices in Afadin's IDR

**Fig 5. Conserved motifs in the Afadin IDR that are predicted to be alpha-helical by AlphaFold correspond with the mapped binding sites of alpha-catenin and actin.** Clustal sequence alignments of the IDRs of vertebrate Afadins. Motifs of 14 amino acids or more that were from 48–88% identical in sequence are highlighted in yellow, or, if the motif was predicted to be alpha-helical by AlphaFold, highlighted in green. Degree of sequence identity in each motif is indicated below the motif. The predicted binding site of alpha-catenin is indicated by blue overlining and overlaps the most N-terminal predicted alpha-helix. The F-actin-binding region identified by Carminati *et al.* is indicated by red overlining and corresponds to the next three predicted alpha-helices. The fragment found by Mandai et al to bind F-actin is indicated by green overlining, the end of the s-Afadin isoform, which does not bind F-actin, is indicated by black overlining, and the "FAB" as referenced in Sakakibara et al 2020 is indicated by cyan overlining.

in the C-terminal FAB region (74% identical from *Drosophila* to *Bradysia*; Fig 6; highlighted in red). Outside the FAB, there were ten motifs in the IDRs, each 14 amino acids or more, that were 48–88% identical in sequence (Fig 6; highlighted in yellow or green). Thus, both insect and vertebrate IDRs contain conserved sequence motifs that have the potential to be protein binding sites.

Recently there were substantial advances in the ability to predict protein structure from primary sequences, using artificial intelligence and machine learning. EMBL's European Bioinformatics Institute has partnered with DeepMind to produce a database of structural predictions of all proteins from multiple model organisms, including *Drosophila* [32, 33]. We used the database to examine the predicted structures of *Drosophila* Cno and human Afadin. In both cases, the full set of known domains, RA1, RA2, FHA, DIL and PDZ, were clearly

# Conservation of protein motifs and predicted alpha-helices in Cno's IDR

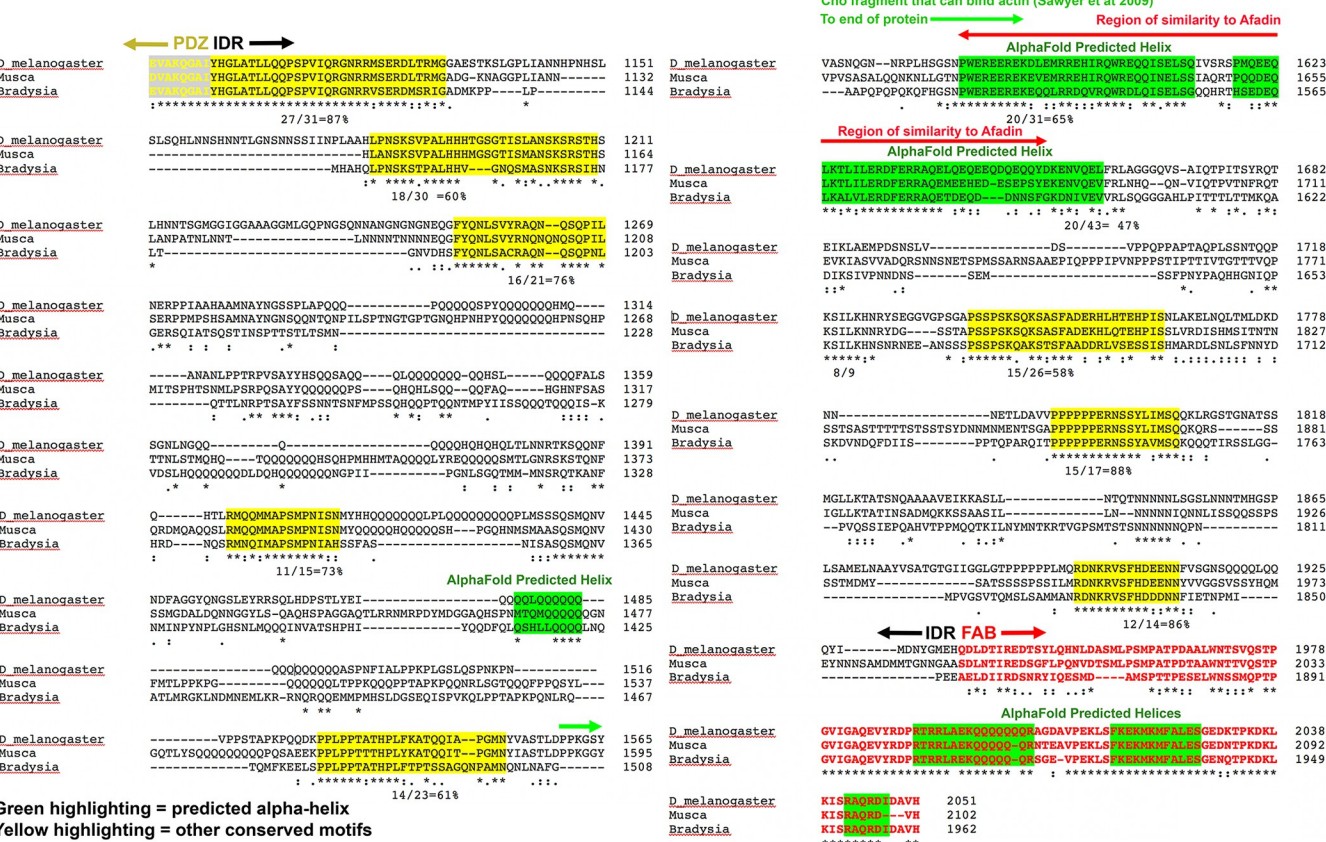

**Fig 6. Conserved motifs in the Cno IDR include several regions that are predicted to be alpha-helical by AlphaFold.** Within the IDR motifs of 14 amino acids or more that were from 48–88% identical in sequence are highlighted in yellow, or, if the motif was predicted to be alpha-helical by AlphaFold, highlighted in green. Degree of sequence identity in each motif is indicated below the motif. The region of Cno found to bind F-actin by Sawyer *et al.*, 2009 is overlined in green. The region of sequence similarity to human Afadin is overlined in red.

included in the predictions (e.g. Fig 7A and 7D green arrows). In some cases, the folded domain predictions included additional structural elements not contained in the Conserved Domain Database (CDD) domain definition.

In contrast, the IDR was largely rendered as an extended chain (e.g., Fig 7A and 7D cyan arrows). However, for both *Drosophila* Cno and human Afadin, there were multiple predicted helices embedded in the IDRs (Fig 7, ends are highlighted by red arrows). In Figs 5 and 6 we highlighted these predicted helices in green in the CLUSTL sequence alignments. The predicted helices included three (*Drosophila* Cno; Fig 7C) or two (human Afadin; Fig 7F and 7G) short helical regions within the FAB region that were well-conserved in insects or vertebrates (Figs 5 and 6, highlighted in red)—two of these overlap between the two phyla. More notable for this discussion, there were additional longer predicted helices within the non-FAB portion of the IDR: three in *Drosophila* Cno (Fig 7A and 7B) and four in human Afadin (Fig 7D and 7E). The four predicted helical regions in human Afadin were included among the most conserved motifs in the IDR, with 69–81% identity, and three of the four form a nearly continuous 173 aa stretch (Figs 5, highlighted in green, 7E). Two of predicted helical regions in *Drosophila* are also closely contiguous (Figs 6, highlighted in green, 7B) and were among the conserved motifs identified in the IDR, with 65% and 47% sequence identity among Diptera, respectively.

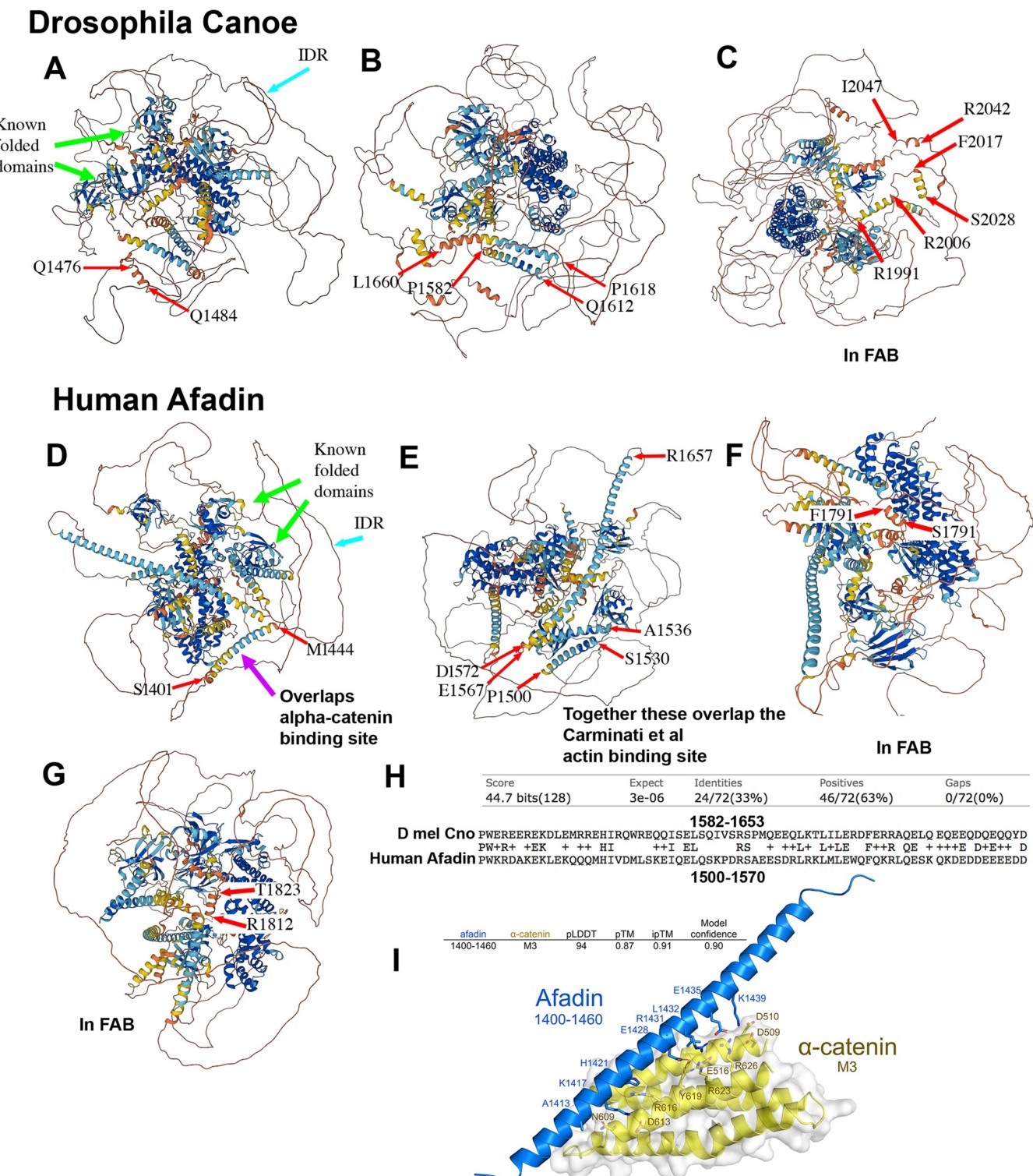

**Fig 7. AlphaFold predicts alpha-helical regions in the IDRs of Cno and Afadin.** Images are from the predicted structures of *Drosophila* Cno and Human Afadin from the AlphaFold Protein Structure Database developed by DeepMind and EMBL-EBI. A-C. Predicted helices in Cno's IDR. Amino acids at the ends of each helix are indicated. D-G.Predicted helices in human Afadin's IDR. H. Blast alignment showing limited sequence identity in some of the predicted alpha-helices in *Drosophila* Cno and human Afadin. I. 3D model of the Afadin/α-catenin complex indicates a conserved dimer interface from human to zebrafish. AlphaFold-Multimer [56]in ColabFold [57] was used to predict heterodimeric structures of the α-catenin-binding region of rat l-Afadin (Uniprot accession: O35889-1, residues 1400–1460) and the M3 domain of mouse alpha-E-catenin (Uniprot accession: P26231, residues 507–631). Models with high

confidence scores (> 0.87) consistently presented a dimer interface between a long α-helix formed by conserved Afadin residues 1403–1455 (blue) and α-catenin M3 (yellow)–the top-ranked model is shown.

These two adjacent motifs in Dipteran IDRs are also conserved in insects outside the Dipterans: for example, there is 56% sequence identity to the beetle *Tribolium* IDR over 70 aas. Most striking, while we detected no sequence similarity in blast searches of the human proteome with the full *Drosophila* IDR, when we searched with the two adjacent helical regions from *Drosophila*, there was a single hit: human Afadin. This region shares 33% identity and 63% sequence similarity over 72 aas (Fig 7H). Together, these data raise the possibility that conserved helical regions in the IDRs play functional roles.

## Published evidence supports the idea that structured conserved motifs in the IDR may be the actin and alpha-catenin binding sites

Structured motifs in some IDRs form the binding sites for protein partners. There are several characterized protein binding sites in the IDR, and we compared these to the conserved motifs identified above, including the subset that are predicted to be structured by AlphaFold. The binding site for alpha-catenin on mammalian Afadin has been defined ([34]; Fig 5, blue double-headed arrow). Strikingly, this region precisely matches the first AlphaFold predicted alpha-helix in the vertebrate IDRs, a motif that is 70% identical between humans and zebrafish (Fig 5). This prompted us to use newly developed structure prediction tools to test the predicted interaction. We used AlphaFold-Multimer (Evans *et al.*, 2021) in ColabFold (Mirdita *et al.*, 2022) to predict heterodimeric structures of the α-catenin-binding region of rat l-afadin (Uniprot accession: O35889-1, residues 1400–1460; [34]) and the M3 domain of mouse α-E-catenin (Uniprot accession: P26231, residues 507–631). Models with high confidence scores (> 0.87) consistently presented a dimer interface between a long α-helix formed by conserved afadin residues 1403–1455 (Fig 7I, blue) and α-catenin M3 (Fig 7I, yellow). Fig 7I displays the top-ranked model which had a predicted confidence of 0.9, with Model accuracy estimates based on the local inter-residue distances (pLDDT), intra-chain arrangements (pTM) and inter-chain interfaces (ipTM) (Evans *et al.*, 2021; Jumper *et al.*, 2021). Strikingly, however, this motif is apparently not conserved in Drosophila—our blast search of the Drosophila proteome and our attempts to use blast to align the motif with the Cno IDR yielded no significant hits, and there are no long conserved motifs in the equivalent region of the Dipteran IDRs (Fig 6, left column bottom). Together, these data suggest this conserved vertebrate Afadin motif is the alpha-catenin binding site, but that this interaction is not conserved in insects.

We next examined whether other conserved motifs might serve as known protein binding sites. Both mammalian Afadin and *Drosophila* Cno contain a C-terminal region that can bind F-actin, but the fragments used in the initial mapping were quite long [4, 19]. More recent reports of the location of the actin-binding site in mammals came to different conclusions. Diagrams recently published by the Takai group (Fig 5 in ref [20]) suggest the FAB region is quite C-terminal, comprising the C-terminal 78 amino acids of l-Afadin (because of apparent alternative splicing between l-Afadin and the isoform we analyzed, in our isoform this region is not entirely contiguous; our Fig 5, cyan arrows). However, attempting to find the data on which this is based proved challenging—the only direct mapping experiments appeared to be the original experiments [19], which implicated a longer region (aa 1609 to the C-terminus in our isoform; Fig 5, green arrow). This was consistent with the observation that a shorter splice product, s-Afadin, which ends at amino acid 1631 in our isoform (Fig 5, black arrow), did not bind actin. However, when Carminati *et al.* more precisely mapped the actin binding site of Afadin [21], their data suggest it maps to a region overlapping but largely distinct from that

defined by the difference between l-Afadin and s-Afadin: aas 1500–1665 in our chosen isoform (Fig 5, red double-headed arrow). Strikingly, the actin-binding region identified by Carminati *et al.* almost precisely matches the region predicted by AlphaFold to be the second, third and fourth predicted alpha-helices in the Afadin IDR (Fig 7E), a region that is overall more than 70% identical between different vertebrates (Fig 5). Intriguingly, this also overlaps the only region of the IDRs that retains detectable sequence conservation between mammals and Drosophila (Fig 7H), a region that includes the second and third predicted alpha-helices in both the Cno and Afadin IDRs (Figs 6, red double-headed arrow, 7B). We discuss the issues raised by the discrepancies in mapping the F-actin binding site more in the Discussion. Together, these data reveal that several of the predicted alpha-helices in the Afadin IDR match predicted binding sites for two of its important partners.

## The zygotic phenotype of the original *cno* mutant that gave the gene its name is paradoxically stronger than that of the null *cno* allele

To test the importance of the IDR as a whole, and of different conserved regions, we will need to carry out functional studies. One potential resource for defining the importance and function of different parts of the Cno protein were classic *cno* mutants, which were isolated due to embryonic lethality and defects in the embryonic body plan. Missense mutations or early stop codons leading to protein truncation can offer insights into the functions of different protein domains (e.g., [35, 36]).

In our lab's analysis of Cno function, we initially sought a protein null mutation as a baseline for protein function. Our standard null allele, $cno^{R2}$, has a premature stop codon quite early in the coding sequence—at amino acid 211 of the 2051 amino acid protein—and does not produce detectable protein [4]. However, when we initially observed its zygotic embryonic lethal phenotype [4], we were surprised at its mildness, as it contrasted with the phenotype of the original *cno* mutant from which the gene got its name [37]. *cno* was first identified in the genetic screen by Nüsslein-Volhard, Wieschaus, and colleagues for embryonic lethal mutations that affect the larval cuticle pattern. Wildtype embryos produce a cuticle that is intact, has a well-developed head skeleton, the result of successful head involution, and which is closed dorsally (Fig 8A). *cno* was one of the genes in which the largest number of alleles were identified—14 alleles [37]. This large target size is consistent with the large size of the *cno* coding sequence. *cno* was named because of the zygotic boat-shaped "dorsal open" cuticle phenotype of at least some of the alleles (e.g., $cno^2$; Fig 8A vs 8D; [38]). This phenotype reflects a failure of dorsal closure, during which sheets of lateral epidermal cells migrate to the dorsal midline, enclosing the embryo in skin— mutants in which dorsal closure failed were named after boats. In contrast, dorsal closure goes to completion in most zygotic null $cno^{R2}$ mutants, and the only defects in most embryos are in head involution, leading to a disrupted head skeleton (Fig 8A vs 8C). Thus, the allele(s) for which the gene got its name have more severe phenotypes than the null mutant—a puzzling and intriguing fact. One possibility is that $cno^2$ encodes a protein that interferes with the function of the wildtype Cno protein put into the egg by the mother, and thus enhances the zygotic mutant phenotype. Consistent with this possibility, there is evidence that some mutant Cno proteins can associate with wildtype Cno [8, 39], and that Afadin can oligomerize [19].

## Maternally contributed Cno persists until late in embryonic morphogenesis, and thus the zygotic null *cno* mutant phenotype is relatively mild

This difference led us to explore the role of maternally-contributed Cno. In most animals, the mother loads supplies into the egg that help drive early embryonic development—in addition

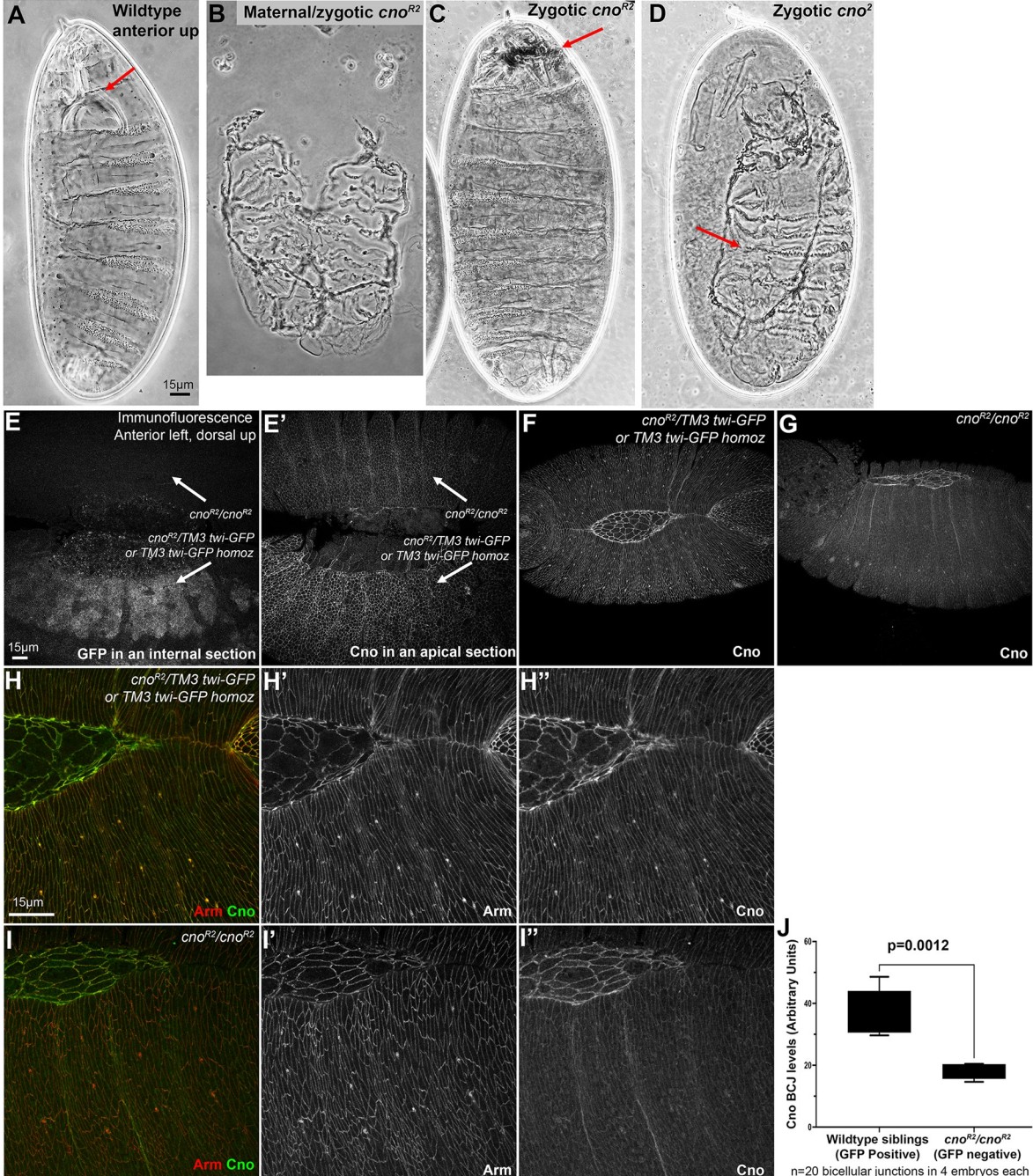

**Fig 8. The zygotic mutant cuticle phenotype of the null allele *cno^R2* is surprisingly less severe than the zygotic phenotype of the "classic" allele *cno^2*.** A-D. Representative cuticles. In this and subsequent Figures, all cuticles are oriented anterior up. A. Wildtype. Head involution is complete, producing a wildtype head skeleton (arrow). Dorsal closure is complete and the epidermis is intact. B. Maternal-zygotic phenotype of the null allele *cno^R2*. Dorsal closure and head involution fail and in the stronger examples like this ventral epidermis is lost. C. Zygotic phenotype of the null allele *cno^R2*. Dorsal closure is completed, and the epidermis is intact. Head involution is disrupted, leading to a fragmented head skeleton (arrow). D. Zygotic phenotype of many embryos of the *cno^2* allele. Both head involution and dorsal closure (arrow) fail, giving the classic "canoe" phenotype. E-I. Immunofluorescence images of Stage 13/14 embryos. In this and all subsequent Figures, unless noted, embryos are oriented with anterior to the left and dorsal side up. E-E'. Use of a Balancer chromosome in which GFP is expressed in the mesoderm under control of the *twist* promotor allows us to identify *cno^R2* homozygotes by lack of GFP expression. F-I. Representative *cno^R2* homozygotes and wildtype or heterozygous siblings, stained to visualize Arm or Cno. Cno accumulates at lower but still detectable levels at cell-cell junctions in *cno^R2* homozygotes. J. Quantification of Cno levels.

to nutritional supplies, these include mRNAs and proteins for the molecular machinery that will power events like chromosome replication and segregation or cell division before activation of the zygotic genome. In some animals, these maternal supplies can last throughout the entirety of embryonic development. The rapidity of *Drosophila* embryonic development, including the exceptional rapid nuclear divisions in the syncytial phase, mean that for many genes, this maternal contribution suffices for the early events of embryonic development—in fact, for some genes, even those that encode proteins with vital roles in embryonic development, the maternal contribution is sufficient for hatching as a viable larva [40].

Thus, the zygotic phenotype of a mutation is often less severe than the phenotype when both maternal and zygotic contributions are removed. While zygotic mutants for our standard null allele *cno^R2* only have defects in head involution, the last major event of morphogenesis, maternal/zygotic mutants that completely lack functional Cno have major defects in most processes of embryonic morphogenesis. Complete loss of Cno disrupts apical constriction and invagination of the mesoderm, the convergent elongation movements that drive germband extension, and the late embryonic collective cell migration events driving dorsal closure and head involution [4, 5]. These mutants also have defects in the integrity of the ventral epidermis, the tissue most sensitive to reductions in cell adhesion. These defects in embryonic morphogenesis are apparent in the larval cuticle secreted at the end of embryonic development. The cuticles of *cno^R2* maternal/zygotic null mutants exhibit defects resulting from failure of dorsal closure and head involution, as well as reduced ventral epidermal integrity (Fig 8A vs 8B; [4]). In contrast, zygotic null *cno^R2* mutants have much milder cuticle defects. In most embryos the epidermis is intact, dorsal closure is complete, and the only morphogenetic event interrupted is head involution, leading to defects in the head skeleton (Fig 8C; [4]).

To assess the degree of maternal Cno contribution directly and quantitatively, and to determine when Cno is depleted in embryonic development, we stained embryonic progeny of a cross of parents heterozygous for the *cno* null mutation, *cno^R2*. We distinguished zygotic mutants using a GFP-marked Balancer chromosome. In late-stage embryos, we could easily identify *cno^R2* homozygotes by their lack of mesodermal expression of GFP under the control of the *twist* promotor (Fig 8E–8E'). In early embryos, Cno accumulation in zygotic mutants remained similar to that seen in their wildtype siblings. It was not until stage 13, as embryos began dorsal closure, that Cno protein became noticeably reduced in *cno^R2* homozygous zygotic mutants relative to heterozygous or wildtype siblings (Fig 8E', 8F vs. 8G, 8H vs 8I). However, Cno remained easily detectable at cell junctions. We quantified the difference by staining the embryos with our C-terminal anti-Cno antibody and imaging both mutant embryos and their wildtype siblings on the same slides using the same microscope settings. We measured pixel intensity at bicellular junctions, subtracting the cytoplasmic background. Overall junctional Cno levels in *cno^R2* homozygous zygotic mutants were reduced to $50.5 \pm 8.0\%$ of those in their wildtype or heterozygous siblings (Fig 8J). This strong persistence of maternal Cno provides a likely explanation for the observation that zygotic mutants only fail during the latest morphogenetic movements.

## A series of classical *cno* mutants provide a potential resource for assessing the function of regions of the IDR

This observation suggested that looking at the molecular lesions in some of the classical *cno* alleles might provide insights into protein function. However, after the 30 years since the Nüsslein-Volhard/Wieschaus screen, only two alleles remained in stock collections—now called *cno^2* and *cno^3*. Fortunately, in the mid-2000's the Gaul and van Aelst labs generated a new series of *cno* alleles on a chromosome carrying an FRT site (the *cno^RX* alleles), allowing the

subsequent generation of maternal and zygotic mutants via the FLP/FRT/Dominant Female Sterile (DFS) approach [41]. Thus these alleles are all on precisely the same genetic background. Two of these, $cno^{R2}$ and $cno^{R10}$, were the alleles we used in our initial analyses of the role of Cno, as both carried early stop codons and appeared to be protein null [4]. However, the others remained uncharacterized. They, along with the two remaining alleles from the Nüsslein-Volhard/Wieschaus screen, provided a potential resource of missense or protein truncating mutants to define the role of the IDR or the folded protein domains.

We thus set out to identify the lesions in all the $cno^{RX}$ alleles and in the two alleles remaining from the Nüsslein-Volhard/Wieschaus screen. Of the 27 $cno^{RX}$ alleles, we still maintained 20. One of these was no longer Balanced, suggesting the $cno$ allele was lost. We verified that the remaining mutants were allelic to $cno$ by assessing complementation for adult viability with the $cno^{R2}$ allele—all failed to complement. Since the $cno$ coding sequence spans more than 44 kb and includes 20 exons, manually amplifying and sequencing the coding exons was daunting. Instead, we performed whole genome sequencing of each heterozygous mutant stock, obtaining 5–7 Gbp of raw sequencing data for each, which provided >25X genome coverage. After mapping the reads to the *Drosophila* genome, we identified the genetic variants across the genome using the freebays [42] and UnifiedGenotyper [43] software packages. We then manually inspected the heterozygous variants identified at the $cno$ locus and filtered out mutations that were in common across all mutants, which were likely variants on the Balancer chromosome. We also prioritized mutations that fell within coding sequences or at splice sites, as these were more likely to be causative.

For 18 of the 21 verified $cno$ alleles, a unique heterozygous change in the coding sequence was observed that was consistent with a deleterious allele. Of these, 14 introduced a premature stop codon, 2 affected conserved splice donor sequences, 1 introduced a 119 nucleotide deletion that results in a frameshift and premature stop codon, and 1 introduced a missense mutation in an amino acid in the FAB region that is conserved among insect Cno relatives. For the splice donor site mutants, we bioinformatically verified that intron inclusion would lead to a frameshift and subsequent stop codon. Each of these alleles was then confirmed by PCR amplification and Sanger sequencing from the heterozygous mutant stock, producing the expected double peak at the site of the mutation (examples are shown in Fig 9A). For the other 3 of the 21 alleles, no lesion was identified in the coding sequence or splice sites. Together, the 18 verified alleles include a set of predicted truncated proteins that span much of the sequence of Cno protein (Fig 9B). Eight alleles have premature stop codons at different positions within the IDR. They thus provided a potential resource for exploring IDR function.

## The zygotic mutant phenotypes of the alleles vary in their severity and those with the strongest phenotypes cluster near the beginning of the IDR

Since the larval cuticle provides a sensitive readout of many of the aspects of embryonic morphogenesis that require cell adhesion and the connection to the cytoskeleton, we used it to compare the phenotypes of our different $cno$ alleles. Together, they offered the chance to assess the strength of mutations leading to truncations across the span of the protein, by comparing their zygotic cuticle phenotypes. We began by validating that our current scoring of cuticle phenotypes matched our previous analysis of the zygotic phenotype of the protein null allele $cno^{R2}$ and the reported phenotype of $cno^2$, the strongest of the $cno$ alleles remaining from the Nüsslein-Volhard/Wieschaus screen [37, 38, 44], from which the gene $cno$ got its name. We scored individual cuticles of dead embryos (n≥100 per genotype), placing each into a category along the spectrum of phenotypes (Fig 10A–10F), using a numerical scoring scheme: 0 = wild-type, 1 = defects in the head skeleton, 2 = failure of head involution, 3 = failure of head

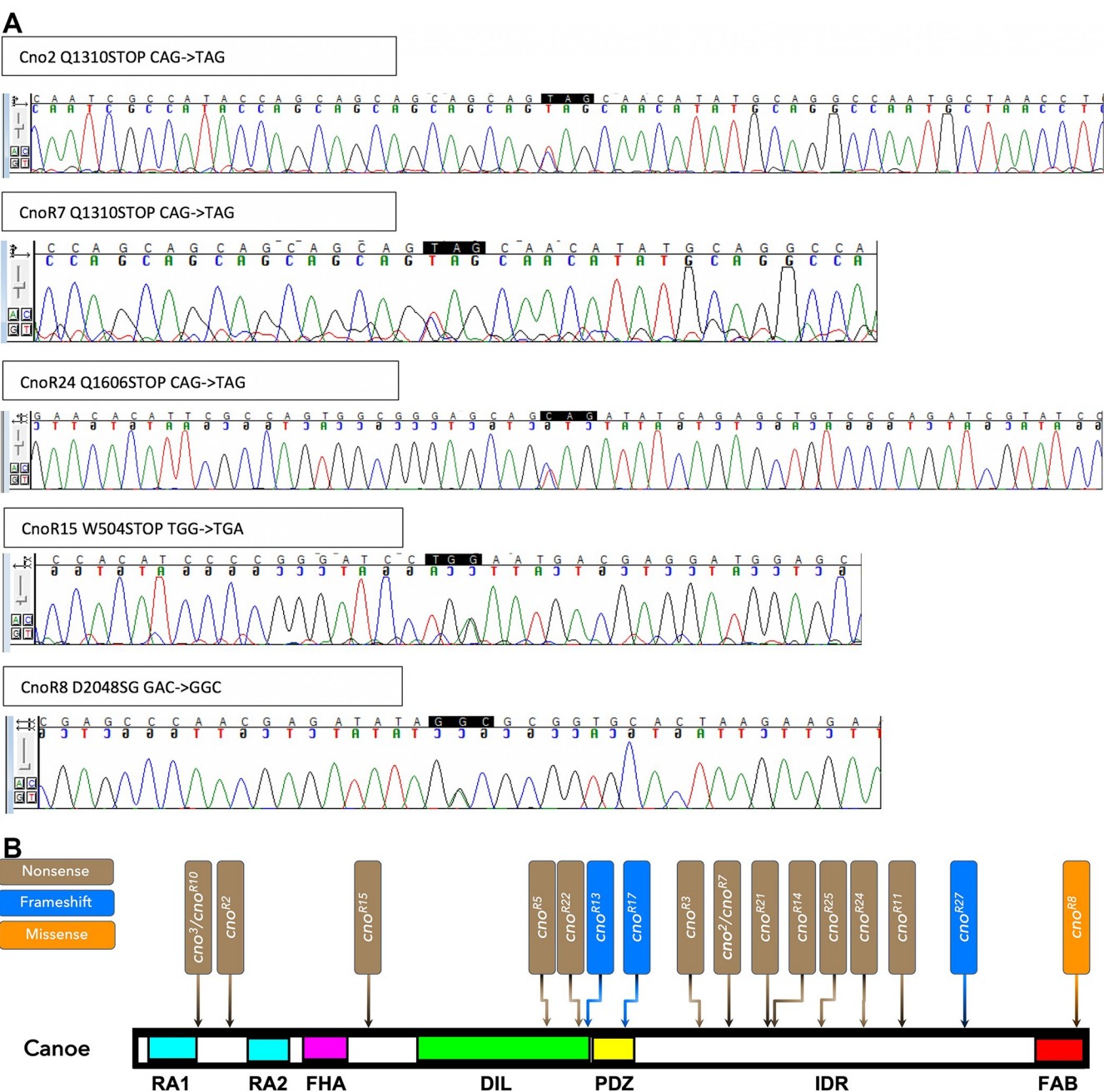

**Fig 9. Our set of *cno* alleles include mutations leading to predicted early stop codons and potential truncated proteins across the span of the Cno protein.** A. Examples of Sanger sequencing confirming the mutations found in whole genome sequencing, as indicated by double peaks in the chromatogram. B. Diagram illustrating the positions and nature of the 18 mutants with a clear lesion in the coding sequence.

involution plus holes in the dorsal or ventral cuticle, 4 = complete failure of both head involution and dorsal closure, or 5 = a fragmented cuticle. We then calculated an average cuticle score for each allele.

We first replicated the surprising difference between the null allele, *cno*$^{R2}$, and one of the original *cno* mutants, *cno*$^2$. 60% of *cno*$^{R2}$ embryos were in the two weakest phenotypic categories, appearing wildtype or with defects in the head skeleton and <8% were in the three

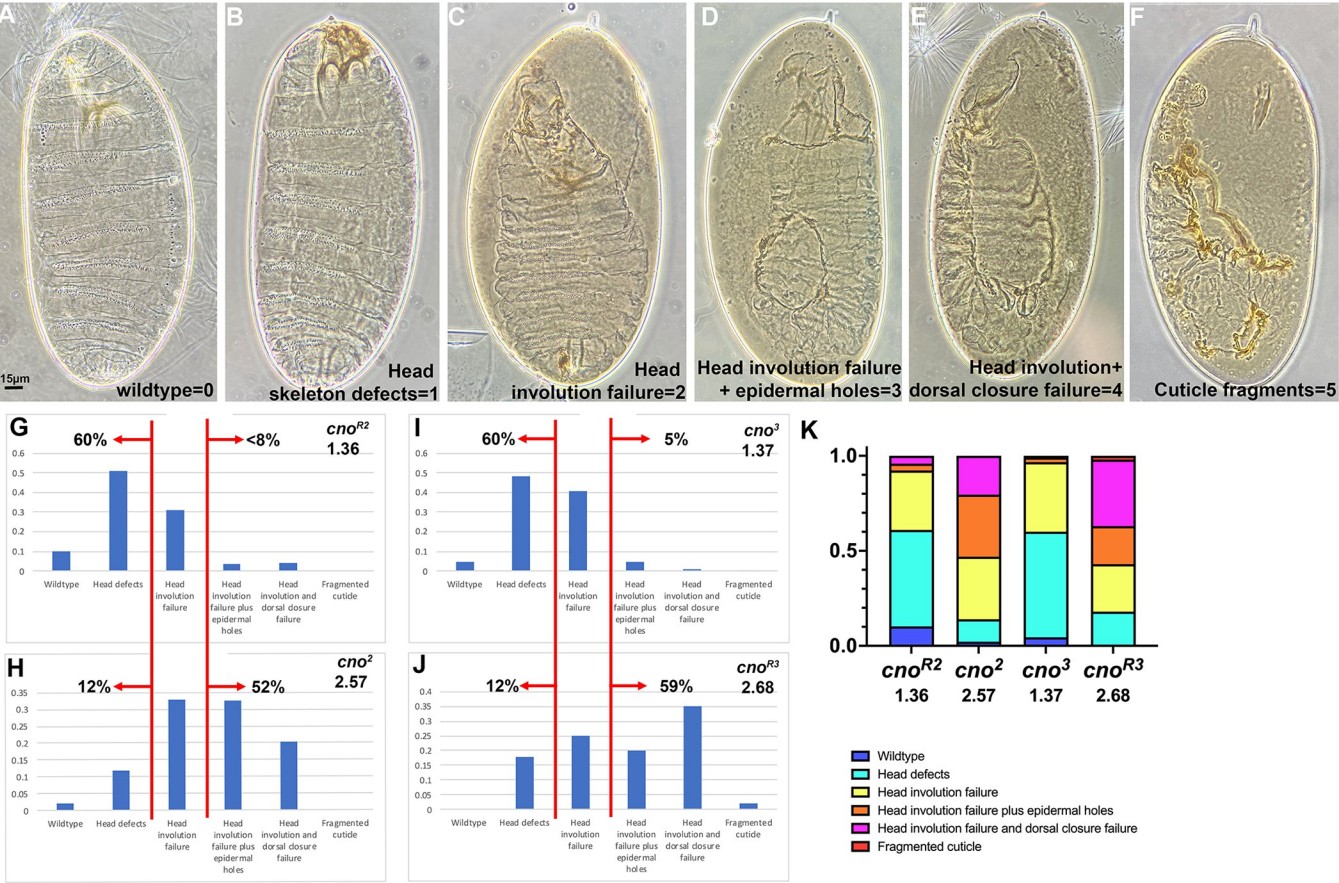

**Fig 10. The cuticle phenotypes of zygotic mutants fall into two broad classes, with the null allele *cno^R2* in the less severe category, and genetic background is not the sole cause.** A-F. Representative cuticles of the six categories used as criteria in scoring, and their numerical equivalents. Anterior up. G, H. Cuticles of the null allele *cno^R2* are on average quite a bit less severe than those of the "classic" allele *cno^2*. I,J. Genetic background is not the sole cause of this difference, as *cno^R3*, which has a strong phenotype, is on the same background as *cno^R2*, while *cno^3*, which has a weaker phenotype, is on a third genetic background. K. Comparisons of the phenotypes of the four *cno* alleles displayed as a 100% cumulative bar chart.

stronger phenotypic categories in which head involution failure was accompanied by holes in the cuticle or failure of dorsal closure (Fig 10G and 10K)—these embryos had an average cuticle score of 1.36 (Table 1). In contrast, only 12% of *cno^2* embryos were in the weakest two phenotypic categories (wildtype or with defects in the head skeleton), while 52% were in the strongest phenotypic categories (head involution failure accompanied by holes in the cuticle or failure of dorsal closure; Fig 10H and 10K)—this yielded an average cuticle score of 2.57 (Table 1).

One potential cause of this difference in phenotype was that these two alleles were isolated on different genetic backgrounds. However, in our collection we had another allele with a very early stop codon like that in *cno^R2*; this was *cno^3*, which is on a different background, as it was isolated in the Nüsslein-Volhard/Wieschaus screen. Like *cno^R2*, *cno^3* had relatively mild phenotypes, with only 5% of embryos in the three stronger phenotypic categories and 60% in the two weakest categories (Fig 10I and 10K; cuticle score 1.37; Table 1). We also had an allele on the *cno^RX* genetic background, *cno^R3*, which has a stop codon early in the IDR, near that seen in *cno^2*. Intriguingly, *cno^R3* also was phenotypically strong, with 59% of the embryos in the three stronger phenotypic categories and only 12% in the two weakest categories (Fig 10J and 10K; cuticle score 2.68; Table 1). These data reveal that genetic background is not the sole basis

**Table 1.  Lesions and phenotypes of *cno* alleles under study.**

| Allele | Lesion | Mutation confirmed by Sanger Sequencing | Zygotic mutant cuticle score | Number of cuticles scored | M/Z mutant Cuticle score |
|--------|--------|------------------------------------------|------------------------------|---------------------------|--------------------------|
| cno[2] | Q1310STOP | Yes | 2.57 | 279 | ND |
| cno[3] | Q140STOP | Yes | 1.37 | 243 | ND |
| **cno[R2]** | K211STOP | NA | 1.36 | 195 | ND |
| cno[R3] | Q1262STOP | Yes | 2.68 | 100 | 3.75 |
| cno[R5] | K833STOP | Yes | 1.56 | 156 | 4.04 |
| cno[R6] | No lesion found | NA | 1.02 | 264 | ND |
| cno[R7] | Q1310STOP | Yes | 1.94 | 183 | ND |
| cno[R8] | D2048G | Yes | 0.77 | 258 | 0.72 |
| cno [R10] | Q140STOP | NA | 1.89 | 197 | 3.93 |
| cno [R11] | R1680STOP | Yes | 1.62 | 250 | ND |
| cno [R13] | Splice N994 frameshift stop donor | Yes | 1.16 | 159 | ND |
| cno [R14] | Q1399STOP | Yes | 1.01 | 186 | ND |
| cno [R15] | W504STOP | Yes | 2.06 | 345 | ND |
| cno [R16] | No lesion found | NA | Not embryonic lethal | NA | ND |
| cno [R17] | Splice R1077 frameshift stop donor | Yes | 2.42 | 188 | ND |
| cno [R21] | Q1389STOP | Yes | 1.51 | 109 | ND |
| cno [R22] | Q992STOP | Yes | 1.1 | 242 | ND |
| cno [R24] | Q1606STOP | Yes | 1.41 | 207 | 3.86 |
| cno [R25] | Q1491Stop | Yes | 1.44 | 152 | ND |
| cno [R26] | No lesion found | NA | 1.57 | 152 | ND |
| cno [R27] | Q1716fs 119 nt deletion | Yes | 1.34 | 209 | ND |

of phenotypic severity differences. In the future, it would be useful to ensure that each allele can be rescued by a transgene carrying wildtype *cno*, to ensure that other mutations on each chromosome are not affecting the phenotype.

Encouraged by this, we completed analysis of the cuticle phenotypes of all the mutants with identified lesions, comparing the position of the early stop codon/probable frameshift and their average zygotic cuticle phenotype (Fig 11A; Table 1). Phenotypes varied (Fig 11B and 11C): one was quite weak (most embryos wildtype or with modest head skeleton defects; *cno*[R8]), many had phenotypes resembling the null allele *cno*[R2] (with head involution failure the predominant phenotype), while a few had many embryos with the classic dorsal open *cno* phenotype (e.g., *cno*[R17]). We were intrigued to see that the lesions in four of the five mutants with the strongest cuticle phenotypes clustered in a particular small region of the protein from the end of the PDZ domain into the first quarter of the IDR (Fig 11A, in which the five strongest

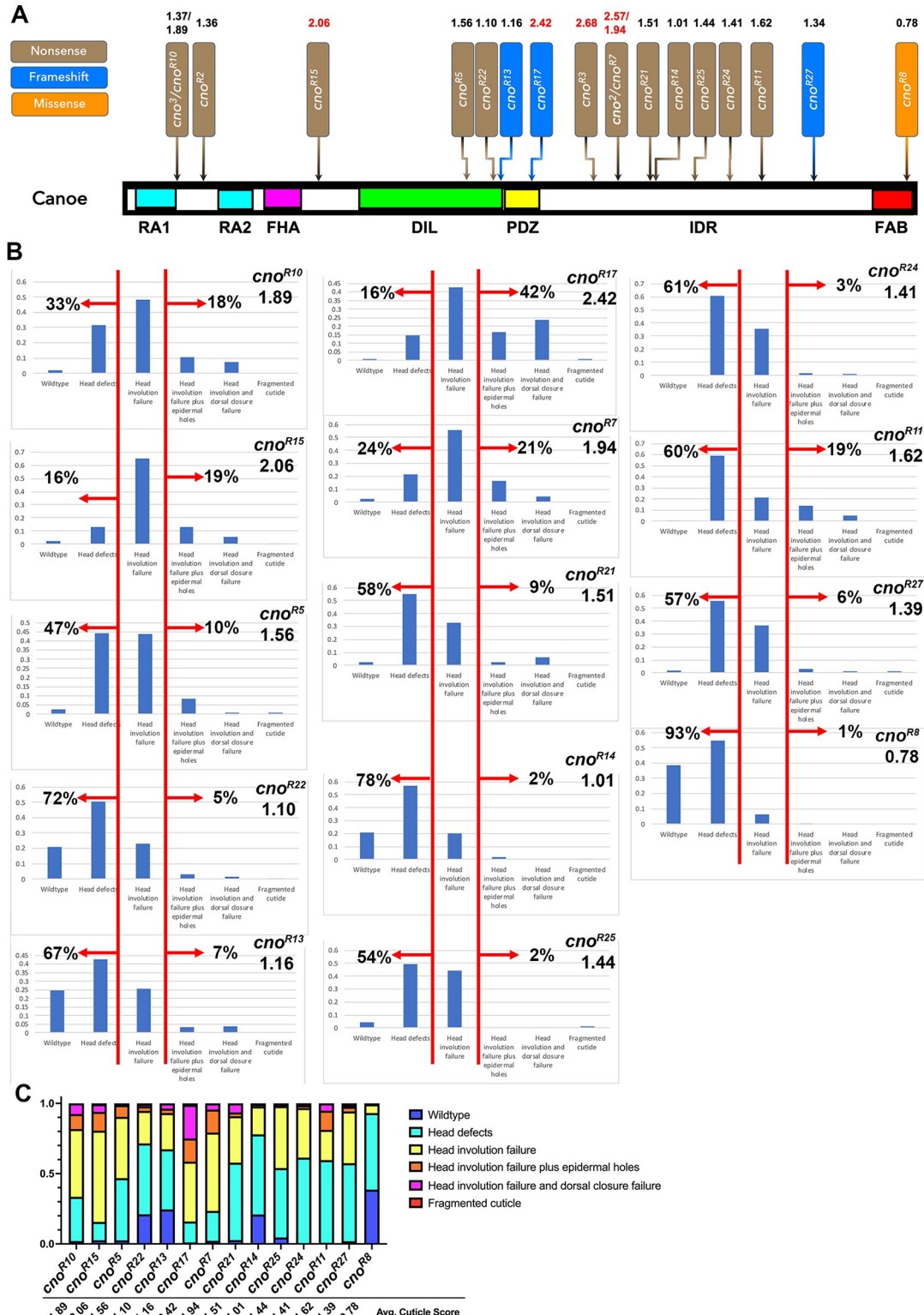

**Fig 11. Four of the five alleles with the strong phenotype result from stop codons near the beginning of the IDR.** A. Positions of the alleles with identified lesions and their "numerical cuticle score". The five strong alleles are highlighted in red. B. Distribution of the cuticle phenotypes of the alleles not presented in Fig 10. C. Comparisons of the phenotypes of these *cno* alleles displayed as a 100% cumulative bar chart.

alleles have their phenotype score highlighted in red). Perhaps not surprisingly, the sole missense mutant, $cno^{R8}$, was one of the weakest. We also analyzed the zygotic mutant phenotype of the three alleles for which we did not find a lesion in the coding sequence. $cno^{R6}$ and $cno^{R26}$ had cuticle phenotypes similar to most of the others, with cuticle scores of 1.02 and 1.57, respectively. $cno^{R16}$ was not embryonic lethal though it did fail to complement the null allele for adult viability. We discuss potential implications of the fact that truncations early in the IDR had more severe phenotypes in the Discussion.

We furthered this analysis of the zygotic phenotypes by comparing the effects on cell shapes and morphogenesis on embryos during and after dorsal closure in one of our putative null alleles with an early stop codon, $cno^{R10}$, with one of our alleles with the strong phenotype, $cno^{R17}$. We distinguished homozygous mutants from heterozygous siblings using a Balancer chromosome expressing GFP in the mesoderm under the control of the *twist* promotor (Fig 12A and 12B bottom right vs top). Homozygous mutant embryos completing dorsal closure had lower but still detectable levels of Cno staining (Fig 12A" vs 12B"), consistent with our analysis above.

During early dorsal closure (stage 13), both mutants had only mild defects, with slightly more persistent segmental grooves (Fig 12C–12E, between red arrows) and somewhat more variability in leading edge cell shape. However, as dorsal closure should have been completed, divergence from wildtype, as well as differences between the two alleles, became more pronounced. All homozygous mutants from both alleles had defects in head involution (Fig 12F vs 12G, 12H and 12J, red arrows; $cno^{R10}$ 14/14 embryos; $cno^{R17}$ 12/12 embryos), consistent with their cuticle phenotypes. Several aspects of the $cno^{R17}$ phenotype were more severe than that of $cno^{R10}$. Most $cno^{R17}$ mutants also exhibited complete failure of dorsal closure, with the gut protruding (Fig 12I, green arrow; 11/13 embryos scored), while this was much less common among $cno^{R10}$ mutants (3/11 embryos scored). A subset of $cno^{R17}$ mutants had holes in the ventral cuticle (Fig 12J, cyan arrow; 3/11 embryos scored) something we did not observe in $cno^{R10}$ mutants (0/15 embryos scored). Once again, these differences in severity at the end of dorsal closure reflect the differences in severity of the cuticle phenotype assessed above. Together, these data are consistent with the idea that alleles with stop codons near the beginning of the IDR have phenotypes more severe than those of the zygotic null alleles.

## Analysis of maternal/zygotic mutants reveals that alleles truncating the coding sequence at different places all lead to very strong loss of function

The strong maternal contribution of Cno means the zygotic phenotype does not reflect the full function of Cno in embryonic development. As noted above, the maternal/zygotic phenotype of the null allele, $cno^{R2}$, is much more severe than the zygotic phenotype, reflecting complete failure of both head involution and dorsal closure and loss of part of the ventral epidermis [4]. To more fully assess the function of alleles truncating the protein at different positions, we used the FLP/FRT Dominant Female Sterile (DFS) approach [41] to generate maternal/zygotic mutants for a subset of our molecularly characterized alleles, and examined their cuticle phenotypes. For several of the alleles, we could not obtain embryos. We suspect additional germline lethal mutations in other genes have accumulated on the right arm of the third chromosome, which becomes homozygosed in this strategy. However, we were able to obtain maternal/zygotic mutants for four truncation alleles, together spanning much of the length of Cno protein—we also generated maternal/zygotic mutants for the missense mutant $cno^{R8}$.

All exhibited 48–65% lethality when females carrying germline clones homozygous for the *cno* mutation were crossed to heterozygous mutant males, as expected for full maternal/zygotic lethality and substantial zygotic rescue, which we previously observed for $cno^{R2}$ [4]. As

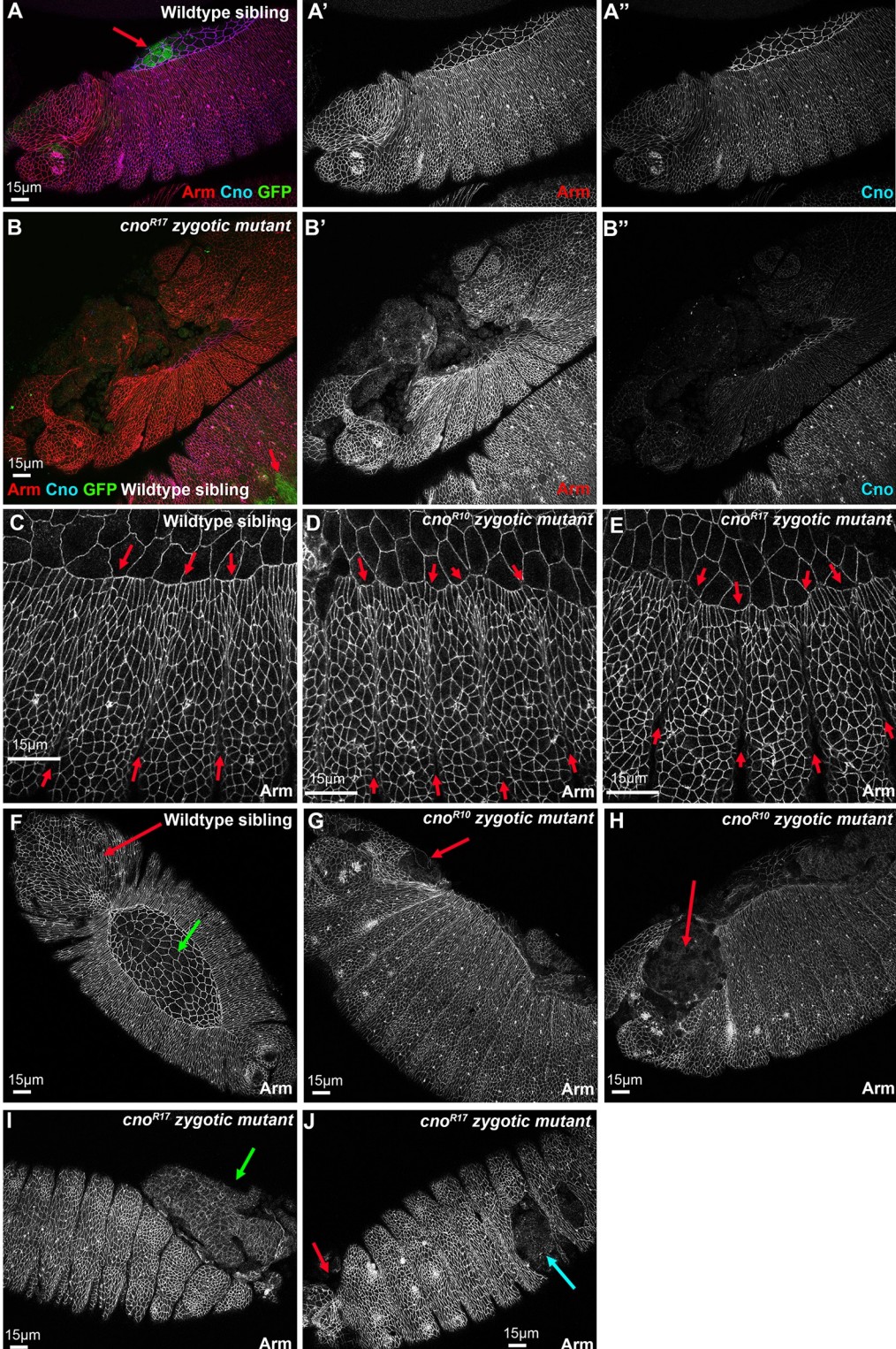

**Fig 12. The strong class of *cno* alleles have more severe effects on morphogenesis than the presumptive null alleles.**
Embryos, anterior left, dorsal side up. Stages 13 (A, C-E) or 14/15 (B, F-J). A,B. GFP expression in the mesoderm from the Balancer chromosome allows us to distinguish Balancer or heterozygous wildtype siblings (A, B bottom left) from homozygous zygotic mutants (B). Zygotic mutants also have strongly reduced Cno staining at AJs. C-E. At stage 13 mutant phenotypes are relatively mild, with more persistent segmental grooves (red arrows) and somewhat more

variable leading edge cell shapes. F. Wildtype sibling showing normal head involution (red arrow) and intact amnioserosa (green arrow). G, H. $cno^{R10}$ zygotic mutant exhibit fully penetrant defects in head involution (red arrows). I, J. $cno^{R17}$ zygotic mutant exhibit fully penetrant defects in head involution (J, red arrow), but also have occasional holes in the ventral cuticle (I, cyan arrow), as well as catastrophic defects in dorsal closure, leading to the gut protruding (I, green arrow).

expected, $cno^{R10}$ –which like $cno^{R2}$ has a very early stop codon–had a maternal/zygotic cuticle phenotype that was much more severe than its zygotic phenotype (Fig 13B vs. 13C, 13I vs 13J, 13O), and which was similar to that we had previously characterized for $cno^{R2}$ [4]. Most $cno^{R10}$ maternal/zygotic mutants had complete failure of dorsal closure and head involution, and many had additional holes in the remaining cuticle. We also generated maternal/zygotic mutants for three additional truncation alleles: $cno^{R5}$, with a stop codon in the Dilute domain (Fig 13D, 13K and 13O), $cno^{R3}$, with a stop codon early in the IDR (Fig 13E, 13L and 13O), and $cno^{R24}$, with a stop codon midway through the IDR (Fig 13F, 13M and 13O). All had similarly strong phenotypes, suggesting all lead to very strong to complete loss-of-function. $cno^{R8}$, the allele with a missense change in the FAB region, was an exception. The maternal/zygotic phenotype of this allele was quite mild, as most mutant embryos had only mild defects in head involution (Fig 13G, 13N and 13O). We discuss the implications of these findings in the Discussion.

We also examined the maternal/zygotic truncation mutant embryos using immunofluorescence to examine morphogenesis as embryos completed dorsal closure. We distinguished maternal/zygotic mutant embryos from zygotically rescued siblings using Cno staining (Fig 14A and 14A'). Maternal/zygotic mutant embryos lost junctional Cno signal and only had occasional signal at segmental grooves that we suspect is background (Fig 14B" vs 14C"). All four truncation alleles exhibited similar very severe phenotypes, with failure to complete dorsal closure before the amnioserosa underwent apoptosis (Fig 14E–14J, magenta arrows), holes in the ventral epidermis (Fig 14D and 14G, cyan arrows), and persistent deep segmental grooves (Fig 14D and 14F–14I, yellow arrows). These are all phenotypes characteristic of very strong loss of maternal/zygotic Cno function [8]. Thus, both cuticle analysis and examination of embryos at the end of dorsal closure revealed that all four truncation alleles lead to very strong loss-of-function.

## Loss of Sdk enhances the *cno* null zygotic phenotype, suggesting genetic background can have effects

Our data above suggest that genetic background alone is unlikely to explain differences in the zygotic phenotype of our different *cno* alleles. However, this did not rule out the possibility that genetic background can affect phenotypic strength. There are many examples of strong genetic interaction, including those involving *cno*. For example, reducing the levels of Cno via RNAi strongly enhances the effects of loss of the ZO-1 homolog *polychaetoid (pyd;* [6] and hypomorphic mutations in *cno* and *pyd* have synergistic effects on embryonic morphogenesis [45]. However, both *cno* and *pyd* are genes where maternal and zygotic loss leads to embryonic lethality. We therefore wondered if mutations that on their own are fully viable and fertile, and thus might be found in the background of an otherwise "wildtype" stock, might enhance the zygotic null phenotype of *cno*.

The gene *sidekick (sdk)* provided a good way to assess this issue. It encodes one of the many proteins in the protein network that link AJs to the cytoskeleton. These proteins vary in their importance—some, like E-cadherin, are essential for adhesion, others, like Cno, play key roles in strengthening AJs as cells change shape and move, while still others play only supporting

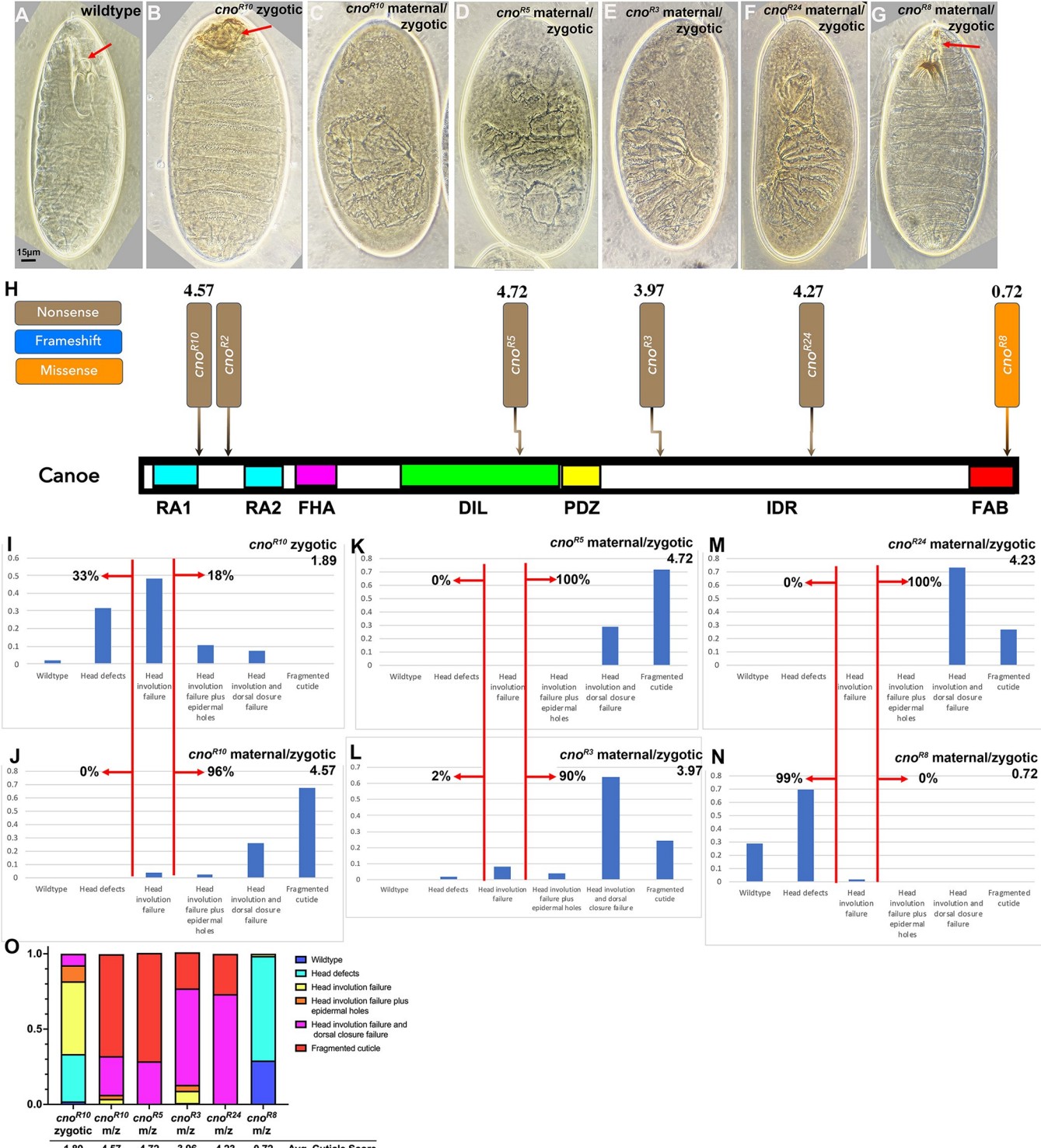

**Fig 13. All the truncation alleles have a very strong maternal/zygotic loss-of-function.** A-G. Representative cuticles of the genotypes indicated, anterior end up. A. Wildtype cuticles have a well-developed head skeleton (arrow). B. Zygotic *cno*[R10] mutants have strong defects in head involution, disrupting the head skeleton (arrow), but are otherwise normal. C. Maternal/zygotic *cno*[R10] mutants exhibit complete failure of head involution and dorsal closure, and many have holes in the ventral cuticle. D-F. Maternal/zygotic *cno*[R5], *cno*[R3], and *cno*[R24] mutants have similar cuticle phenotypes. G. In contrast, while maternal/zygotic *cno*[R8] mutants die as embryos, their cuticles are wildtype or have mild defects in the head skeleton (arrow). H. Diagram showing location of the early stop codons or missense mutation in the alleles examined here and their average cuticle scores. I-N. Distribution of the cuticle phenotypes of the alleles presented. O. Comparisons of the phenotypes of these *cno* alleles displayed as a 100% cumulative bar chart.

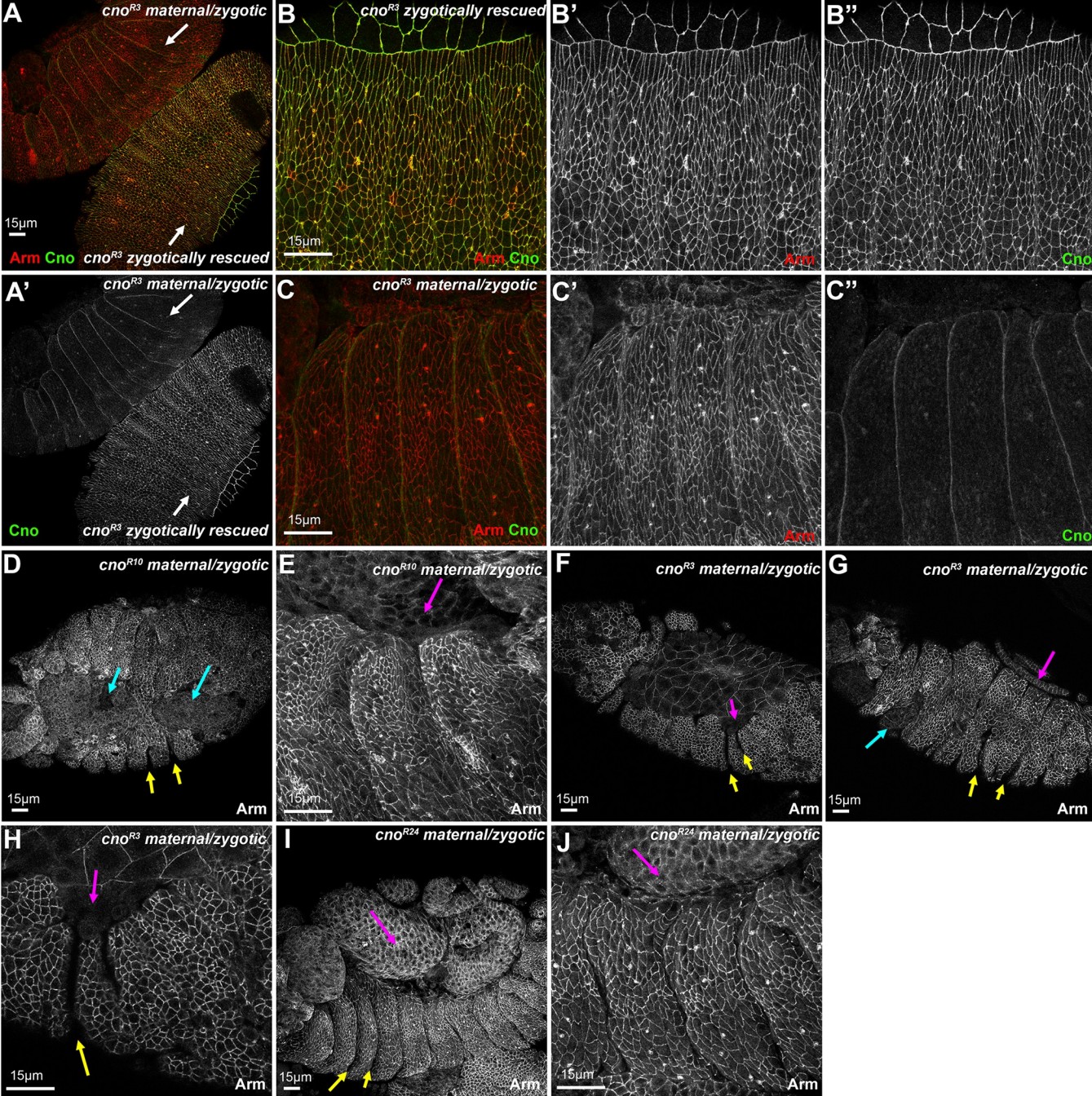

**Fig 14. The three truncation alleles analyzed all disrupt dorsal closure, ventral epidermal integrity and segmental groove retraction.** Embryos, stages 14–15. Anterior to the left. A. Side-by-side maternal/zygotic *cno*[R3] mutant and zygotically-rescued sibling. They are easily distinguished by the restoration of junctional Cno staining. B,C. Closeups of the embryos in A. In maternal/zygotic mutants the junctional Cno signal is lost. D-J. Representative stage 14 or stage 15 embryos from the indicated genotypes. For all four genotypes, maternal/zygotic mutants exhibit dorsal closure failure (magenta arrows), defects in ventral epidermal integrity (cyan arrows) and persistent deep segmental grooves (yellow arrows).

roles. Sdk is one of the latter: *sdk* null mutants are viable and fertile, with defects in adult eye development and neural circuit wiring [46–49]. *sdk* null mutant embryos survive and hatch, albeit with subtle defects in cell shape change and in junctional integrity at tricellular junctions [48]. We thus compared the phenotypes of *sdk* null mutants, *cno*[R2] zygotic mutants, and *cno*[R2]

zygotic mutants that were also maternally and zygotically *sdk* null. We verified that $sdk^{MB5054}$ maternal/zygotic null mutants were viable (4% embryonic lethality (n = 350 embryos), similar to our wildtype stocks. Crosses of $cno^{R2}$/ + parents led to 28% embryonic lethality (n = 608 embryos), as expected from complete embryonic lethality of the 25% who are homozygous mutants and full viability of heterozygotes. In contrast, when we crossed $sdk^{MB5054}$; $cno^{R2}$/ + parents, we observed 49% lethality (n = 445 embryos), suggesting some heterozygous embryos die.

To get a clearer view of potential enhancement of the $cno^{R2}$ zygotic morphogenetic pheno-type, we examined cuticles, using the scoring criteria from above. Most $cno^{R2}$ zygotic mutants had mild to moderate defects in head involution, with 11% in the stronger phenotypic categories in which head involution failure was accompanied by holes in the cuticle or failure of dorsal closure (Fig 15A and 15C), similar to what we previously observed [4]. In contrast, the cuticle phenotypes of the progeny of $sdk^{MB5054}$; $cno^{R2}$/ + parents were substantially more severe, with 50% in the most severe phenotypic categories (Fig 15B and 15C).

We also examined embryos during dorsal closure. Consistent with our earlier work and what we presented above, $cno^{R2}$ zygotic mutants have somewhat deeper segmental groves

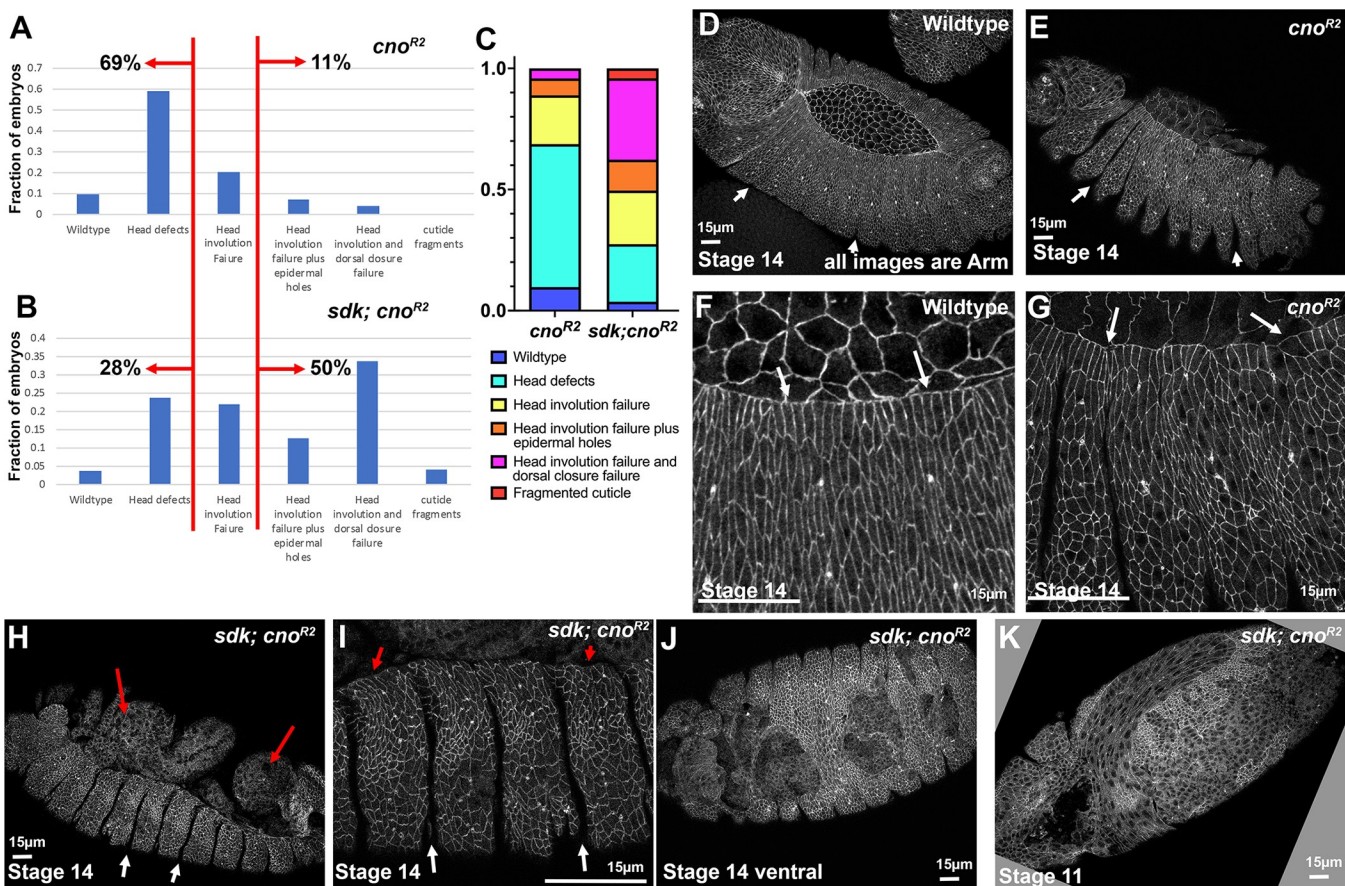

**Fig 15. Loss of Sdk enhances the zygotic phenotype of the null allele $cno^{R2}$.** A,B. Cuticle phenotypes of the null allele $cno^{R2}$ versus those of $sdk$; $cno^{R2}$ mutants. C. Comparisons of the phenotypes of these genotypes displayed as a 100% cumulative bar chart. D-J. Stage 14 embryos stained for Arm, lateral view unless noted. D, F. Wildtype. Dorsal closure and head involution are proceeding. Segmental grooves are no longer deep (D, arrows), and have receded from the leading edge (F, arrows), where cell shapes are relatively uniform. E,G. $cno^{R2}$ zygotic mutant. Segmental grooves remain deep (E, arrows) and leading-edge cell shapes are less uniform (G, arrows). H,I. $sdk$; $cno^{R2}$ mutant. Segmental grooves are very deep (white arrows). Dorsal closure has failed, exposing internal tissues (H, red arrows), and cell shapes at the leading edge are less elongated along the dorsal-ventral axis (I, red arrows). J. Ventral view of a $sdk$; $cno^{R2}$ mutant revealing disruption of the ventral epidermis. K. Stage 11 $sdk$; $cno^{R2}$ mutant with disruptions of the ventral epidermis.

(Fig 15D vs 15E, arrows) and occasional hyper-constricted or hyper-elongated cells at the leading edge (Fig 15F vs 15G, arrows). In contrast, many *sdk^MB5054*; *cno^R2* mutants exhibited failure of dorsal closure, with the amnioserosa having undergone apoptosis before closure was completed (Fig 15H, red arrows). Segmental grooves were very persistent (Fig 15H and 15I white arrows), and leading-edge cell elongation was disrupted (Fig 15I, red arrows). In a subset of the embryos, holes were seen in the ventral epidermis, both during closure (Fig 15J) and even earlier (Fig 15K). These defects are fully consistent with the more severe cuticle phenotypes. Thus, although *sdk* loss does not lead to lethality or substantially disrupt embryonic morphogenesis, it can significantly enhance the zygotic null phenotype of *cno*, and thus genetic background can also play a role.

## Discussion

Our long-term goal is to define the molecular mechanisms that link cell-cell junctions to the cytoskeleton, forging connections that are flexible enough to allow dramatic cell shape changes and movements, yet also robust enough to prevent tearing tissues apart. Mammalian Afadin and its *Drosophila* homolog Cno are critical to this linkage, and we have used Cno as a model to define how complex multidomain proteins contribute to this process. We began by dissecting the function of some of Cno's folded domains, with surprising results [8], but now are turning toward examining the function of the intrinsically disordered region (IDR) that comprises almost half of the protein. IDRs have recently come into focus for their roles in the assembly of diverse multiprotein complexes (Banani *et al*., 2017; Posey *et al*., 2018; Bondos *et al*., 2022), but we have little information about the roles of the IDRs of Cno or Afadin. Here we examine the evolution of these IDRs and begin to define IDR function, using a set of classical *cno* alleles.

### The IDRs of Cno and Afadin share very few commonalities

Despite the ~600 million years separating them from their common ancestor [50], Cno and Afadin share identical domain architectures—each has two Rap1-binding RA domains, predicted FHA and DIL domains, a PDZ domain that can bind to transmembrane junctional proteins, followed by a long IDR ending in a relatively conserved sequence that has been referred to as the FAB region at the C-terminus. During the time since their divergence, the folded domains in mammalian and insect Cno/Afadin have remained conserved, though the degree of conservation varies between different domains. The RA1 and PDZ domains are most conserved, remaining 74% and 71% identical in *Drosophila* versus human, while the other domains are substantially less conserved: 54%, 48%, and 42% for the RA2, DIL, and FHA domains, respectively. It is intriguing that the degree of conservation does not necessarily predict functional importance—mutational analysis revealed that while the RA domains are collectively almost essential for all Cno functions, the PDZ domain is virtually dispensable in the lab, playing only a supporting role despite its strong conservation [8]. It will be interesting in the future to define the function of the other folded domains.

It is not surprising that the IDRs diverged more rapidly between mammalian Afadin and *Drosophila* Cno, as this was observed in other proteins [13]. However, the extent of the difference was dramatic in multiple ways. The IDRs differ in length (705 aas vs 835 aas), and while we previously noted some sequence similarity in the region referred to as the FAB, outside this region our reciprocal Blast searches with the full IDR detected no detectable similarity between human Afadin and *Drosophila* Cno. In many other cases examined, while IDR sequence is not conserved, other molecular features are shared among IDRs in proteins with similar functions, such as enrichment of particular amino acids or net charge (Zarin *et al*., 2019). However, both

of these properties also differ substantially between human Afadin and *Drosophila* Cno. These dramatic differences in sequence and other properties led us to look at divergence at shorter evolutionary time scales.

### Both IDRs contain motifs conserved across shorter evolutional times scales, some of which align with predicted alpha-helical regions that may serve as alpha-catenin and F-actin binding sites

IDRs that diverge in overall amino acid sequence sometimes retain short peptide motifs that serve as binding sites for protein partners [13]. We previously noted the FAB region as a potential example [8], as it was the only IDR sequence clearly conserved between human and *Drosophila* Afadin/Cno. AlphaFold predicts 2–3 alpha-helices in the most conserved C-terminal ~60 amino acids of the FAB region, two of which overlap between flies and mammals. The FAB region is one of the sequences whose function we interrogated in Cno. It is not essential but plays an important supporting role in ensuring robust connections between adherens junctions under tension [8]. This region also plays a functional role in mammalian Afadin, with its deletion slowing the reassembly of adherens and tight junctions [22]. However, as we discuss below, it remains unclear if this region directly interacts with F-actin, as originally proposed.

Outside the FAB, we found little apparent sequence similarity between fly and mammalian IDRs. To identify conserved motifs, we compared IDRs over shorter evolutionary time scales. The result was striking. In vertebrates, the Afadin IDR retains nearly 70% identity overall from human to zebrafish (Figs 2 and 5), thus making it almost as well-conserved as the FHA domain (75% identical). Multiple well-conserved motifs were apparent, with some quite lengthy. For example, the 151 amino acids immediately following the PDZ domain are 74% identical across the vertebrates we examined, and the central region of the IDR contains three consecutive motifs that also are more than 70% identical. We found that four of these conserved motifs correspond to the four regions in the IDR predicted by AlphaFold to form alpha-helices. Intriguingly, the most N-terminal of these overlaps precisely with the mapped binding site on Afadin of alpha-catenin [34], which suggested this helical motif in the IDR interacts with alpha-catenin—our structural modeling strongly supports this. Since alpha-catenin is a critical player in sensing and transducing tension at AJs [2], it will be important in the future to test whether its interaction with Afadin affects its conformation, dimerization state, and binding to other alpha-catenin-binding partners, such as beta-catenin, vinculin, and F-actin. Mutational analysis of this motif in cultured mammalian cells would be one route forward.

The next three predicted helices, which follow closely on one another, almost precisely overlap with the F-actin binding site identified by Carminati *et al.* [21], who used co-sedimentation with polymerized F-actin as their assay. This match is quite striking (Fig 5). The F-actin binding site mapped by Carminati *et al.* does not, however, correspond with the region the Takai lab defined as the F-actin binding (FAB) region, nomenclature we followed in our own earlier analysis. This prompted us to re-examine the published evidence underlying the FAB designation. The initial mapping of the F-actin binding region of Afadin used radioactively-labeled F-actin in a blot overlay assay [19]. This implicated the C-terminal 199 amino acids of the rat Afadin isoform they call l-Afadin as both necessary and sufficient for F-actin binding in this assay. This region only modestly overlaps with the region mapped by Carminati *et al.*—the overlap only includes part of the third of the three predicted long alpha-helices (Fig 5). More puzzling, the Takai lab's most recent publications define the FAB more restrictively, now only including the most C-terminal ~90 amino acids [20, 22]. We could not identify any newly published data supporting the idea that this more restricted region binds actin in vitro,

although it is required for timely re-assembly of adherens junctions in cultured cells [22]. Based on these analyses, we suspect the central set of alpha-helices in the Afadin IDR represent one F-actin binding site, but it is possible there are two regions of Afadin that can bind F-actin —this remains to be determined.

Our analysis revealed that the insect IDR diverged much more rapidly than that in vertebrates (Fig 6). While conservation remains relatively strong within the *Drosophila* genus (75% identity between *D. melanogaster* and *D. virilis*), conservation drops off rapidly relative to other Dipteran insects outside the genus, with 50% identity to the IDR of the housefly *Musca* and only 37% identity between *Drosophila* and the fungus gnat *Bradysia*. Conserved motifs exist, but there are fewer, and they are less extensive than those in the vertebrate IDRs. Alpha-Fold predicts two long alpha-helical regions in the center of the Drosophila IDR, which align with two of the conserved motifs. Intriguingly, these include a region similar in sequence to the predicted alpha-helices defined as an F-actin binding site in Afadin by Carminati et al (Fig 5). This region was included in the fragment of *Drosophila* Cno that can co-sediment with F-actin [4]. It will be important to directly test whether this set of conserved motifs in Cno binds actin. In contrast, there is no sequence in the *Drosophila* IDR similar to the vertebrate alpha-catenin binding site observed in Afadin, and a blast search with the human motif returned no significant similarity in insects. In fact, there are no long conserved motifs N-terminal to the two predicted alpha-helices that comprise the potential F-actin binding site in *Drosophila*. However, mutations altering the M-region of Drosophila alpha-catenin alter the recruitment of Cno to tricellular junctions [55], consistent with some sort of interaction, direct or indirect. Thus, it will be important to determine if *Drosophila* Cno directly interacts with alpha-catenin, and to define the mechanisms by which alpha-catenin regulates Cno recruitment to AJs under tension. Taken together, this analysis reveals conserved motifs, one of which may be shared by *Drosophila* and mammals, whose function can now be examined in vivo, as we discuss below.

### Our set of *cno* alleles are consistent with the idea that the IDR is important for normal Cno function, and that proteins truncated early in the IDR may have dominant-negative effects

Mutational analysis can reveal how the different parts of the multidomain Cno protein contribute to its function. Protein null alleles provided the baseline for full loss of function [4, 5]. We then began to delete individual domains or regions of the protein to assess their contributions. In our initial analyses, we removed the RA domains, the PDZ domain and the "FAB" region [8]. As expected, removing the RA domains, which provide a means by which the small GTPase Rap1 can activate Cno [39, 51], severely reduced protein function. However, the other two mutants proved surprising. We had hypothesized that the PDZ domain and FAB region together provided the binding sites by which Cno connected transmembrane junctional proteins including E-cadherin to the actin cytoskeleton. However, to our surprise, neither region was essential for viability. More sensitive assays revealed that both are required for full reinforcement of AJs when force is exerted on them during embryonic morphogenesis. These data opened the possibility that other regions of the protein might play more important roles.

The *cno* alleles assessed here provide an alternate approach to define the function of different domains/regions, particularly the IDR. As we detail below, our zygotic mutant data indicate that prematurely truncated proteins may have a dominant-negative effect on wildtype Cno, suggesting that IDR-truncated proteins may escape from nonsense-mediate mRNA decay in the zygotic mutants. However, maternal/zygotic mutants for these same truncated alleles exhibit a severe phenotype comparable to that of the null mutant, suggesting that proteins lacking the IDR retain little function on their own. This strongly suggests an essential role of IDR.

Maternal/zygotic mutants, in which the only protein in the embryo is the mutant protein, provide definitive assessment of the function of an allele. We assessed four alleles with early stop codons spread across the protein. $cno^{R10}$, with a very early stop codon, provides the baseline for complete loss of function. Its cuticle phenotype matched that of our canonical null allele $cno^{R2}$, and mutant embryos had complete failure of head involution and dorsal closure, with the integrity of the ventral epidermis–which is most sensitive to reduction in cell adhesion [52]–often disrupted. Intriguingly, mutants with predicted truncations in the DIL domain ($cno^{R5}$), early in the IDR ($cno^{R3}$), or truncated in the first of the long alpha-helices in Cno's IDR predicted by AlphaFold ($cno^{R24}$) all also had similarly strong phenotypes, consistent with strong or complete loss-of-function. In contrast, $cno\Delta FAB$ [8], which ends just 330 amino acids C-terminal to the predicted end of $cno^{R24}$, is zygotically viable, and most maternal zygotic mutants also survive embryogenesis. This difference is consistent with the idea that the C-terminal region of the IDR, which includes the alpha-catenin and internal F-actin binding sites, is critical for protein function.

There is a caveat to drawing definitive conclusions from these data, however. Many early stop codons trigger nonsense-mediated mRNA decay [53]. If all our alleles exhibited strong mRNA decay, then they would not encode proteins, and we could not draw inferences from the position of the stop codon. However, nonsense variants can escape from mRNA decay via a variety of mechanisms [35, 54], and escape is more common with alleles in which the stop codon is closer to the end of the gene. For example, in previous analysis of mutations in Drosophila *armadillo*, the beta-catenin homolog, we found that many mutations with early stop codons, most of which were not in the terminal exon, encode detectable proteins, and some accumulate sufficient protein to provide substantial remaining function [35]. In that case, we could detect these truncated proteins as the antibody recognized an N-terminal epitope. Unfortunately, the epitope for our Cno antibody is a C-terminal protein sequence [4], and this part of Cno sequence is predicted to be completely or nearly completely absent in all of our truncation alleles, with the possible exception of $cno^{R27}$. Thus, we were unable to use immunoblotting to assess whether the predicted truncated proteins are made.

However, the apparently antimorphic phenotype of alleles with stop codons early in the IDR is consistent with them encoding stable protein. Because of the strong maternal contribution of *cno* mRNA and protein, and the persistence of maternal Cno to late morphogenesis, the zygotic phenotype is quite sensitive to what are likely to be small differences in protein function. The null allele only exhibits disruption in the latest event of morphogenesis, head involution. Further, $cno\Delta RA$, in which maternal/zygotic mutants have a very strong loss-of-function, is zygotically embryonic viable [8]. Most of our alleles were zygotically embryonic lethal with cuticle defects very similar to those of the null allele, consistent with strong loss-of-function. However, a subset of the alleles had a stronger phenotype, with a large fraction of embryos with defects in dorsal closure. Four of the five strongest alleles, including the three strongest ($cno^2$, $cno^{R3}$, and $cno^{R17}$), have truncations at or near the beginning of the IDR. This is consistent with the idea that these alleles encode truncated proteins that interact with and interfere with the function of the diminishing levels of maternal Cno. Both Drosophila Cno and mammalian Afadin can dimerize or oligomerize [19, 39].

We further tested the strength of these possible dominant-negative effects by examining effects on heterozygous viability. To do so, *we out crossed females from four alleles Balanced over the TM3 Balancer to wildtype males, and counted Balanced and unbalanced progeny. We carried this out at 18 degrees and 25 degrees. The four alleles we used were*: *1)* cno$^{R2}$ [our canonical null allele with a early stop codon and a "weak" phenotype, 2+3) $cno^2$ and $cno^{R17}$, two alleles with stop codons near the beginning of the IDR and the two alleles with the strongest

**Table 2. Tests of heterozygous viability.**

| | Relative Viability vs Balancer Chromosome | |
|---|---|---|
| | 18°C | 25°C |
| *cno^{R2}*/TM3 x WT | 54% | 53% |
| *cno^2*/TM3 x WT | 54% | 49% |
| *cno^{R17}*/TM3 x WT | 51% | 54% |
| *cno^{R8}*/TM3 x WT | 49% | 57% |

phenotypes 4) *cno^{R8}*, our only missense allele and the allele with the weakest phenotype. The data are in Table 2.

If these strong alleles had a strong dominant-negative effect, we predicted that their heterozygous viability would be lower than that of the null and potentially that of the weak allele *cno^{R8}*. However, there were no consistent differences suggesting strong effects of the two alleles with truncations early in the IDR. At 18 degrees *cno^{R8}* had the lowest viability and *cno^2* viability was identical to that of *cno^{R2}*. At 25 degrees, while *cno^2* viability was lower, the viability of *cno^{R17}* was higher than that of the null allele. Thus, none of the alleles has a strong dominant-negative effect—we thus suspect the effects we see are due to the very low levels of maternal zygotic Cno protein late in embryonic development, making the phenotype susceptible to subtle disruptions of maternal protein function. To fully define whether proteins truncated in the IDR are dominant negative, it will be important to engineer alleles that produce truncated proteins without the caveat of early stop codons, as we did with cnoΔFAB, to test this hypothesis.

*cno^{R8}* does not carry an early stop codon, but instead has a missense change, aspartic acid to glycine, in an amino acid conserved in *Drosophila*, the fungus gnat *Bradysia*, and in the more distantly related butterfly *Heliconius*. There is a conserved glutamic acid in a similar position in vertebrate Afadin. We were initially surprised a missense change in the FAB region was lethal, since deleting the FAB is not lethal [8]. However, Yu and Zallen found that mutating a different conserved residue in the FAB, tyrosine 1987, to alanine also reduced Cno function [3]. Perhaps these missense proteins dimerize with wildtype Cno and reduce its function. Our analysis of maternal/zygotic *cno^{R8}* mutants suggest that this mutant protein retains quite a bit of function, more consistent with our analysis of *cnoΔFAB*.

We also explored the potential role of genetic background in the phenotype of zygotic mutants. With respect to our mutations encoding premature stop codons, we saw no strong correlation between genetic background and phenotypic strength. There were mutants from the "R allele" set with phenotypes in both the "weak" and "strong" categories, and the two alleles we scored from other backgrounds included one in each category. We did note small differences in strength between independent mutations with the same lesion, which may reflect background. However, our work exploring genetic interactions between *sdk* and *cno* revealed that it is possible to have non-lethal mutations present in a genetic background, which strongly enhance the *cno* zygotic phenotype. This genetic interaction also reinforces the idea that AJ: cytoskeletal connections are not linear, but involve a network of interactions. Some protein-linkages may not be essential in isolation, but may reinforce other connections, enhancing robustness (e.g. [8, 55]).

Together this bioinformatic and mutational analysis sets the stage for our future work. It will now be important to use our CRISPR-based platform to introduce back into the endogenous *cno* gene mutations affecting the IDR, beginning with a clean deletion of the entire IDR. It will also be of interest to replace the *Drosophila* IDR with that of mammalian Afadin, to test conservation of function. We can then begin to test the potential roles of conserved motifs within the IDR, both genetically and biochemically.

## Methods

### Sequence alignments and other sequence analyses

FastA protein sequences were downloaded from NCBI and aligned with the Multiple Sequence Alignment program Clustal Omega [25, 26] from EMBL/EBI. Asterisks indicate consensus sequence identity, colons positions with conservation between groups of strongly similar properties, and periods conservation between groups of weakly similar properties as below—definitions can be found at [https://www.ebi.ac.uk/seqdb/confluence/display/JDSAT/Bioinformatics +Tools+FAQ#BioinformaticsToolsFAQ-WhatdoconsensussymbolsrepresentinaMultipleSequ enceAlignment?]. The sequences highlighted as domains were defined as follows: 1) RA1 and RA2 were delineated according the CDD database (cd01782 and cd01781, respectively. 2) The FHA domain was defined using a published NMR structure = pdb|1WLN|A Chain A, Afadin. 3) The DIL domain was defined by comparison of the AlphaFold prediction of Cno/Afadin and the solved Myosin V structure. 4) the PDZ domain was defined by the published structure of the Cno PDZ domain [8].

### Structural predictions

Predicted alpha-helices depicted in Fig 7 were derived from the AlphaFold Protein Structure Database Developed by DeepMind and EMBL-EBI [32, 33]. To generate the 3D model of the afadin/α-catenin complex, AlphaFold-Multimer [56] in ColabFold [57] was used to predict heterodimeric structures of the α-catenin-binding region of rat l-afadin (Uniprot accession: O35889-1, residues 1400–1460; [34]) and the M3 domain of mouse α-E-catenin (Uniprot accession: P26231, residues 507–631). Model accuracy estimates are provided based on the local inter-residue distances (pLDDT), intra-chain arrangements (pTM) and inter-chain interfaces (ipTM) ([33, 56]. Model confidence = $0.8 \cdot \text{ipTM} + 0.2 \cdot \text{pTM}$ [56].

### Whole genome sequencing and Sanger sequencing verification

We isolated genomic DNA from each heterozygous Balanced stock and whole genome sequencing of the *cno* mutants was performed by BGI Genomics, using their "DNBseq plant and animal WGRS" service. This yielded 50–70 million reads (5–7 Gbp) for each sample. This equates to 25-40X genome coverage. Reads were mapped to the *Drosophila melanogaster* genome (build dm6.36) using bbmap ([58], the duplicate reads were removed with samtools [59], and the genome variants were called using the freebays [42] and UnifiedGenotyper (DePristo) algorithms with the default parameters. The resulting SNP calls were combined and filtered (—filterExpression "QUAL < 199 || DP < 2") using the GATK package [43] prior to SNP annotation with the snpEff software [60]. SNPs for each mutant were called relative to the reference genome, except for the *cno*[2] and *cno*[3] mutants, which were called relative to their respective background chromosomes: the *ru cu ca* background chromosome for *cno*[2] and or a *hh* mutant isolated on the same *st e* background for *cno*[3]. Heterozygous missense or nonsense mutations at the *cno* locus were selected for further analysis. First, mutations found in all *cno* mutant lines were attributed to the TM3 balancer chromosome and discarded. Next, nonsense mutations were prioritized, as these lesions are predicted to result in the strong loss-of-function phenotypes that are expected. Finally, the top candidate *cno* lesion for each allele was verified by Sanger sequencing of the genomic DNA, as follows: Genomic DNA was extracted from 1 male and 1 female adult fly of each allele by crushing both in 10mM Tris pH 8.0, 1mM EDTA, 25mM NaCl, and 10 ug Proteinase K, then heating to 37˚C for 30 minutes and 95˚C for 2 minutes. Regions of interest for each allelic mutation were PCR amplified with primers generating 200–2000 bp long products using Phusion Hot Start II DNA Polymerase (Thermo

Fisher F549S), using the recommended reaction conditions. These were sent for Sanger sequencing by Eton Biosciences—usually internal primers not used for amplification were used for sequencing. Polymorphisms were detected using Sequencher.

## Cuticle analysis

All analysis was done at 25˚C. Cuticle preparation was performed according to [61]. In brief, embryos were aligned on agar plates and allowed to develop fully. Unhatched embryos were nutated in 50% bleach to remove the chorion membrane. After nutation, the embryos were washed in 0.1% Triton X-100 and mounted on glass slides in 1:1 Hoyer's medium/lactic acid. The glass slides were incubated at 60˚C for 48 h and then stored at room temperature. Photos were taken on a Nikon Optiphot 2 using a 20x Ph2 DL Phase objective.

## Embryo fixation and immunofluorescence

Flies were crossed in cages over apple juice agar plates with yeast paste and left to lay eggs for 4–18 h before collection. Our method of embryo collection, embryo fixation, and embryo staining was previously described by [39]. Briefly, for heat fixation: We removed the chorion membrane by nutating embryos in 50% bleach solution for 4 minutes. Afterward, the embryos were washed three times in 0.03% Triton X-100 with 68 mM NaCl, and then fixed in 95˚C Triton salt solution (0.03% Triton X-100 with 68 mM NaCl and 8 mM ethylene glycol-bis(2-aminoethylether)-$N,N,N',N'$-tetraacetic acid [EGTA]) for 10 s. We fast-cooled samples by adding ice-cold Triton salt solution and placing on ice for at least 30 min. We removed the vitelline membrane by vigorous shaking in 1:1 heptane/methanol solution. The embryos were again washed thrice with 95% methanol/5% EGTA and stored in 95% methanol/5% EGTA for up to 24 h at −20˚C before staining. Before staining, the heat-fixed embryos were washed three times with 5% normal goat serum/0.1% saponin in phosphate-buffered saline (PBS) (PBSS-NGS). Embryos were then blocked in the same solution for 1 h. The embryos were then incubated with the primary antibodies overnight at 4˚C, washed three times with PBSS-NGS, and incubated with secondary antibodies overnight at 4˚C. Both primary and secondary antibodies were diluted in 1% bovine serum albumin/0.1% saponin in PBS (PBSS-BSA), and the dilutions used are listed in Table 3. After the secondary antibody incubation, we washed three times with PBSS-NGS and stored embryos in 50% glycerol until mounted on glass slides using a homemade Gelvatol solution (recipe from the University of Pittsburgh's Center for Biological Imaging).

**Image acquisition and analysis.** All images were obtained from fixed embryos. We imaged on a Carl Zeiss LSM 880 confocal laser-scanning microscope. Images were captured on the 40×/1.3 NA Plan-Apochromat oil objective. Brightness and contrast were fine-tuned

**Table 3. Antibodies and probes used in this study.**

| Antibodies | Species | Dilution | Source |
|---|---|---|---|
| Anti-Cno | Rabbit IgG | 1:1000 | Sawyer *et al.*, 2009 |
| Anti-Armadillo (N2 7A1) | Mouse IgG$_{2a}$ | 1:100 | Developmental Studies Hybridoma Bank |
| Anti-GFP (NBP2-50034) | Chicken IgY | 1:500 | Novus Biologicals |
| **Secondary Antibodies** | | | |
| Alexa Fluor 488 Plus | Anti-Rabbit IgG | 1:1000 | Invitrogen |
| Alexa Fluor 647 | Anti-Mouse IgG$_{2a}$ | 1:1000 | Invitrogen |
| DyLight 550 | Anti-Chicken IgY | 1:1000 | Invitrogen |

using both ZEN 2009 software and after image processing in Adobe Photoshop. To capture the enrichment of proteins at the adherens junctions, we created MIPs with tools from ImageJ (National Institutes of Health). MIPs for Cno antigen intensity were created from Z-stacks ($1024 \times 1024$ pixels) through the dorsolateral epidermis of stage 13–15 embryos. The Z-stack images were captured using a digital zoom of 1.8 and a step size of 0.2 μm.

### Antigen intensity comparison

We used MIPs spanning a 1.0 μm apical section of AJs from embryos between stages 13 and 15 to compare antigen intensity in heterozygous versus homozygous $cno^{R2}$ embryos. Using Fiji/ImageJ, a six-panel grid was first generated over the area of interest to ensure randomized selection of cell borders. Next, lines of fixed length and width were drawn at a mixture of anterior/posterior and dorsal/ventral junctions of cells along the lateral sheets (below cells of the leading edge); circles of the same width were drawn in the cytoplasm of these same cells. Close-up images (1.8X zoom with 40X-oil objective) were acquired for each embryo. For each close-up image, 20 lines and 12 circles were drawn and subsequently analyzed for average pixel intensity. With these data, two values were calculated: average line intensity and average circle intensity. Finally, to calculate a single value for pixel intensity at junctions, we treated pixilation within the circles as background signal and, thus, subtracted the average circle intensity from the average line intensity. This process was repeated for 5 GFP-positive embryos (2 close-up images per embryo; total of 10 close up-images) and 6 GFP-negative embryos (2 close-up images for 4 embryos and 1 close-up image for 2 embryos; total of 10 close-up images). To calculate Cno junctional intensity in GFP-negative relative to GFP-positive embryos, we divided the average of the 10 GFP-negative junctional intensity values by the average of the 10 GFP-positive junctional intensity values (and multiplied by 100%). To quantify the average intensity difference, we needed to find the standard deviation of the ratio between the average GFP-negative intensity and the average GFP-positive intensity. Given that standard deviation is defined as the square root of the variance, we defined a function, $f$, as the ratio X/Y, where X and Y represent the sets of GFP-negative and GFP-positive values, respectively. We then used a Taylor's Expansion to find the variance of $f$. This was relatively easy to do since we already had the means ($\mu_X$ and $\mu_Y$) and variances ($Var[X]$ and $Var[Y]$) of the individual sample sets: $Var\left[\frac{X}{Y}\right] \approx$

$$Var\left[\frac{\mu_X}{\mu_Y} + \frac{1}{\mu_Y}(X - \mu_X) - \frac{\mu_X}{\mu_Y^2}(Y - \mu_Y)\right] = Var\left[\frac{1}{\mu_Y}X - \frac{\mu_X}{\mu_Y^2}Y\right] = \frac{1}{\mu_Y^2}Var[X] + \frac{\mu_X^2}{\mu_Y^4}Var[Y] - 2\frac{\mu_X}{\mu_Y^3}Cov[X, Y].$$

### Acknowledgments

We are grateful to Ulrike Gaul, Benjamin Boettner, Linda Van Aelst, and colleagues, who generated and generously shared the R series *cno* alleles, Maik Bischof and Corbin Jensen for advice and assistance with statistical analysis, Kevin Slep for advice about protein structure and the use of AlphaFold data, to Kristi Yow for training new lab members in *Drosophila* genetics and cell biology, to Kia Perez-Vale for advice on Sanger sequencing, to Corbin Jensen, to Emily McParland, and other Peifer lab members for helpful advice and discussions, to Flybase for information on different Cno isoforms, to the Developmental Studies Hybridoma Bank (DSHB) for antibodies, and the Bloomington and Kyoto Drosophila Stock Centers for fly stocks, and the three reviewers for helpful suggestions. We thank Tony Perdue of the Biology Imaging Center for confocal imaging advice and support. We are grateful to Zuhayr Alam, Victoria Williams, and Anna Dibattista for technical assistance.

## Author Contributions

**Conceptualization:** Noah J. Gurley, Rachel A. Szymanski, Robert H. Dowen, Mark Peifer.

**Data curation:** Noah J. Gurley, Rachel A. Szymanski, Robert H. Dowen, Mark Peifer.

**Formal analysis:** Noah J. Gurley, Rachel A. Szymanski, Robert H. Dowen, Mark Peifer.

**Funding acquisition:** Robert H. Dowen, Mark Peifer.

**Investigation:** Noah J. Gurley, Rachel A. Szymanski, Robert H. Dowen, T. Amber Butcher, Noboru Ishiyama, Mark Peifer.

**Methodology:** Noah J. Gurley, Rachel A. Szymanski, Robert H. Dowen, T. Amber Butcher, Noboru Ishiyama, Mark Peifer.

**Project administration:** Mark Peifer.

**Supervision:** Mark Peifer.

**Writing – original draft:** Noah J. Gurley, Rachel A. Szymanski, Robert H. Dowen, Mark Peifer.

**Writing – review & editing:** Noah J. Gurley, Rachel A. Szymanski, Robert H. Dowen, T. Amber Butcher, Noboru Ishiyama, Mark Peifer.

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
