## [Decision Letter · Decision Letter 0]

12 May 2023

PONE-D-23-06606Exploring the evolution and function of Canoe’s intrinsically disordered region in linking cell-cell junctions to the cytoskeleton during embryonic morphogenesisPLOS ONE

Dear Dr. Peifer,

Thank you for submitting your manuscript to PLOS ONE. After careful consideration, we feel that it has merit but does not fully meet PLOS ONE’s publication criteria as it currently stands. Therefore, we invite you to submit a revised version of the manuscript that addresses the points raised during the review process.

We look forward to receiving your revised manuscript.

Kind regards,

Giovanni Messina

Academic Editor

PLOS ONE

“We are grateful to Ulrike Gaul, Benjamin Boettner, Linda Van Aelst, and colleagues, who generated and generously shared the R series cno alleles, Maik Bischof and Corbin Jensen for advice and assistance with statistical analysis, Kevin Slep for advice about protein structure, to Kristi Yow for training new lab members in Drosophila genetics and cell biology, to Kia Perez-Vale for advice on Sanger sequencing, to Corbin Jensen, to Emily McParland, and other Peifer lab members for helpful advice and discussions, to Flybase for information on different Cno isoforms, to the Developmental Studies Hybridoma Bank (DSHB) for antibodies, and the Bloomington and Kyoto Drosophila Stock Centers for fly stocks. We thank Tony Perdue of the Biology Imaging Center for confocal imaging advice and support. We are grateful to Zuhayr Alam, Victoria Williams, and Anna Dibattista for technical assistance.  This work was supported by NIH R35 GM118096 to M.P. Work in the Dowen lab is supported by NIH R35 GM137985.  The authors declare no competing financial interests.”

“This work was supported by National Institutes of Health R35 GM118096 to M.P

https://www.nih.gov/

Work in the Dowen lab is supported by National Institutes of Health R35 GM137985 to R.H.D.

https://www.nih.gov/

Reviewers' comments:

Reviewer's Responses to Questions

**Comments to the Author**

1. Is the manuscript technically sound, and do the data support the conclusions?

Reviewer #1: Yes

Reviewer #2: Yes

Reviewer #3: Yes

2. Has the statistical analysis been performed appropriately and rigorously? 

Reviewer #1: Yes

Reviewer #2: Yes

Reviewer #3: Yes

3. Have the authors made all data underlying the findings in their manuscript fully available?

Reviewer #1: Yes

Reviewer #2: Yes

Reviewer #3: Yes

4. Is the manuscript presented in an intelligible fashion and written in standard English?

Reviewer #1: Yes

Reviewer #2: Yes

Reviewer #3: Yes

5. Review Comments to the Author

Reviewer #1: This manuscript attempts to identify the role of IDRs in Canoe and Afadin by performing an indepth structure analysis along with experiments of mutants in vivo. The work in the manuscript is carried out in a rigorous manner with good controls and enough n values along with the relevant statistics. I recommend the manuscript for publication with a few minor additions suggested below.

The authors find that the IDRs diverged more rapidly as compared to the well defined domains.

One caveat in the study is whether the truncated canoe protein is expressed in the mutants and that was not ascertainable due to the absence of a suitable antibody.

In general the analysis of the different mutants is done well, just one analysis which would have been nice to add is the lethality of the heterozygous mutants themselves and an analysis of heteroallelic combinations to further emphasize the complementation between domains of the mutants. This is not essential but a discussion on this complementation will help understand the function of different domains. This will also help impart an important function to the IDRs.

Since the orientation of the embryos is different directions, it is hard to decipher the phenotypes being described in the figures. I request the authors to as far as possible choose sample images in the same orientation.

Is it possible that the severe defect seen in the canoe mutants in the IDR domain occurs due to partial dominant negative defects. Heterozygous allele embryonic lethality at different temperatures would have been useful to assess this effect.

Also is it known if these mutants can all be rescued by addition of a duplication of the locus or a transgene with the endogenous promoter? This will help delineate the impact of off target mutations in the alleles on the phenotypes.

Reviewer #2: In this manuscript, the authors have extensively investigated the Drosophila Canoe protein and its vertebrate ortholog Afadin using bioinformatic and genetic analyses. The paper is well written and the data are well presented.

Reviewer #3: Gurley et al. investigated the evolutional conservation and function of the intrinsically disordered region (IDR) of Canoe/Afadin in this paper. Using bioinformatics, they found that each of Canoe and Afadin has multiple sequence motifs that are conserved over shorter evolutionary periods. Analysis of zygotic and maternal/zygotic mutants of canoe suggested that IDR is important for Canoe's protein function. These results shed new light on a relatively unexplored region of Canoe/Afadin and provide a new starting point of future research.

Minor points and suggestions

1. Discussion pp. 37-38, lines. 850-865

The logic supporting the conclusion that IDR is important for Canoe function is not easy to follow. This section of the Discussion (and related part) should be rewritten so that the conclusion is convincingly acceptable.

My understanding is as follows:

Zygotic mutant data indicate that prematurely truncated proteins have a dominant-negative effect, suggesting that IDR-truncated proteins may escape from nonsense-mediate mRNA decay in the zygotic mutants. This suggests that these molecules are also expressed in the maternal/zygotic mutants but the mutants exhibit a severe phenotype comparable to that of the null mutant, strongly suggesting an essential role of IDR.

2. Figs. 9G-J, 10B, 12I-N, 14A-B

It is difficult to compare differences between mutants in phenotypic severity at a glance. I would suggest that the authors use a stacked vertical bar chart (100% cumulative bar chart).

3. Fig. 6I

Using AlphaFold and AlphaFold-Multimer, it is predicted that the conserved region of Afadin (1403-1455) forms an alpha-helix and interacts with alpha-E-catenin M3. Since alpha-E-catenin is an important tension transducer of adherens junctions, it would be useful to discuss whether its interaction with Afadin affects its conformation, dimerization state, and binding to other alpha-E-catenin-binding partners (such as beta-catenin, vinculin, and F-actin).

Typographical error

p. 26, l.582-583 seems to be repetitive.

"comparing one of our putative null alleles with an early stop codon, cnoR10, with one of our alleles with the alleles with the strong phenotype, cnoR17."

6. PLOS authors have the option to publish the peer review history of their article (what does this mean?). If published, this will include your full peer review and any attached files.

Reviewer #1: No

Reviewer #2: No

Reviewer #3: No

---

## [Author Response · Author response to Decision Letter 0]

12 Jul 2023

Response to reviews: PONE-D-23-06606

We are grateful to the reviewers for their generally positive response and thoughtful suggestions regarding our manuscript. We have followed their suggestions, addressing all of the issues by adding new data and making the suggested changes in the text and Figures. We carried out the suggested analysis of heterozygous effects on viability suggested by Reviewer 1, added the stacked graph displays of phenotypes suggested by Reviewer 3, and revised the text and Figure presentation as suggested. Figure 7,8,9,10,11,12,13,and 14 were all revised, following the suggestions of the reviewers. We attached a marked-up version of the text with the changes highlighted in red. These revisions strengthened and extended our work and we are grateful for their suggestions. Our responses to each point are below in italics.

Reviewers' comments:

Reviewer's Responses to Questions

Comments to the Author

1. Is the manuscript technically sound, and do the data support the conclusions?

Reviewer #1: Yes

Reviewer #2: Yes

Reviewer #3: Yes

2. Has the statistical analysis been performed appropriately and rigorously? 

Reviewer #1: Yes

Reviewer #2: Yes

Reviewer #3: Yes

3. Have the authors made all data underlying the findings in their manuscript fully available?

Reviewer #1: Yes

Reviewer #2: Yes

Reviewer #3: Yes

4. Is the manuscript presented in an intelligible fashion and written in standard English?

Reviewer #1: Yes

Reviewer #2: Yes

Reviewer #3: Yes

5. Review Comments to the Author

Reviewer comments are included in their entirety, and Responses are in italics

Reviewer #1: This manuscript attempts to identify the role of IDRs in Canoe and Afadin by performing an indepth structure analysis along with experiments of mutants in vivo. The work in the manuscript is carried out in a rigorous manner with good controls and enough n values along with the relevant statistics. I recommend the manuscript for publication with a few minor additions suggested below.

The authors find that the IDRs diverged more rapidly as compared to the well defined domains.

One caveat in the study is whether the truncated canoe protein is expressed in the mutants and that was not ascertainable due to the absence of a suitable antibody.

In general the analysis of the different mutants is done well, just one analysis which would have been nice to add is the lethality of the heterozygous mutants themselves and an analysis of heteroallelic combinations to further emphasize the complementation between domains of the mutants. This is not essential but a discussion on this complementation will help understand the function of different domains. This will also help impart an important function to the IDRs.

We are grateful for this favorable assessment and thoughtful suggestions. We have made sure the important caveat you note is clearly stated in the Discussion. 

Since the orientation of the embryos is different directions, it is hard to decipher the phenotypes being described in the figures. I request the authors to as far as possible choose sample images in the same orientation.

Our apologies for our lack of clarity in this regard, giving us the opportunity to clarify image orientation throughout. It is the convention in our field to orient immunofluorescence images of embryos anterior to the left, and, unless a different view is needed, oriented dorsal up. We have 1) verified that all of our images fit this convention 2) clearly labeled the first Figure where we show these images, and 3) Edited each of the Figure legends to explicitly state the orientation. Our labs practice and that of many of our colleagues for cuticle images is different—these are oriented anterior up, to facilitate comparison between wildtype and mutants. For cuticles we have also clearly labeled the first Figure where we show these images, and edited each of the relevant Figure legends to explicitly state the orientation.

Is it possible that the severe defect seen in the canoe mutants in the IDR domain occurs due to partial dominant negative defects. Heterozygous allele embryonic lethality at different temperatures would have been useful to assess this effect.

This was a good suggestion and we have carried out the suggested experiment. We out crossed females from four alleles Balanced over the TM3 Balancer to wildtype males, and counted Balanced and unbalanced progeny. We carried this out at 18 degrees and 25 degrees. The four alleles we used were: 1) cnoR2 [our canonical null allele with a early stop codon and a “weak” phenotype, 2+3) cno2 and cnoR17, two alleles with stop codons near the beginning of the IDR and the two alleles with the strongest phenotypes 4) cnoR8, our only missense allele and the allele with the weakest phenotype. The data are below.

 Relative viability versus Balancer Chromosome

 18° 25° 

cnoR2/TM3 x WT 54% 53% 

cno2/TM3 x WT 54% 49% 

cnoR17/TM3 x WT 51% 54% 

cnoR8/TM3 x WT 49% 57% 

If these strong alleles had a strong dominant-negative effect, we predicted that their heterozygous viability would be lower than that of the null and potentially that of the weak allele cnoR8. However, as you will see, there were no consistent differences suggesting strong effects of the two alleles with truncations early in the IDR. At 18 degrees cnoR8 had the lowest viability and cno2 viability was identical to that of cnoR2. At 25 degrees, while cno2 viability was lower, the viability of cnoR17 was higher than that of the null allele. Thus, none of the alleles has a strong dominant-negative effect—we thus suspect the effects we see are due to the very low levels of maternal zygotic Cno protein late in embryonic development, making the phenotype susceptible to subtle disruptions of maternal protein function. To fully define whether proteins truncated in the IDR are dominant negative, it will be important to engineer alleles that produce truncated proteins without the caveat of early stop codons, as we did with cno∆FAB, to test this hypothesis. We now have added this data and the accompanying text to the relevant section of the Discussion. 

Also is it known if these mutants can all be rescued by addition of a duplication of the locus or a transgene with the endogenous promoter? This will help delineate the impact of off target mutations in the alleles on the phenotypes.

Given the number of alleles involved, this would be a very substantial amount of work. Instead, we have noted the caveat as follows: : “In the future, it would be useful to ensure that each allele can be rescued by a transgene carrying wildtype cno, to ensure that other mutations on each chromosome are not affecting the phenotype.” We did ensure that the lethality of all alleles was due to a lesion in cno by verifying that all are lethal over the null allele, cnoR2. We also now clarify that most of the alleles were generated on the same genetic background, and added the following caveat

Reviewer #2: In this manuscript, the authors have extensively investigated the Drosophila Canoe protein and its vertebrate ortholog Afadin using bioinformatic and genetic analyses. The paper is well written and the data are well presented.

We are grateful for this favorable assessment.

Reviewer #3: Gurley et al. investigated the evolutional conservation and function of the intrinsically disordered region (IDR) of Canoe/Afadin in this paper. Using bioinformatics, they found that each of Canoe and Afadin has multiple sequence motifs that are conserved over shorter evolutionary periods. Analysis of zygotic and maternal/zygotic mutants of canoe suggested that IDR is important for Canoe's protein function. These results shed new light on a relatively unexplored region of Canoe/Afadin and provide a new starting point of future research.

We are grateful for this favorable assessment and thoughtful suggestions.

Minor points and suggestions

1. Discussion pp. 37-38, lines. 850-865

The logic supporting the conclusion that IDR is important for Canoe function is not easy to follow. This section of the Discussion (and related part) should be rewritten so that the conclusion is convincingly acceptable.

My understanding is as follows:

Zygotic mutant data indicate that prematurely truncated proteins have a dominant-negative effect, suggesting that IDR-truncated proteins may escape from nonsense-mediate mRNA decay in the zygotic mutants. This suggests that these molecules are also expressed in the maternal/zygotic mutants but the mutants exhibit a severe phenotype comparable to that of the null mutant, strongly suggesting an essential role of IDR.

Thank you for this very accurate assessment and concrete suggestion. In attempting to avoid making unwarranted conclusions, we had made this section so convoluted that our major points and their caveats were lost. We have completely revised this section of the Discussion, putting a summary like that above at the beginning of the section, and then shortening and clarifying the more detailed discussion that follows. 

2. Figs. 9G-J, 10B, 12I-N, 14A-B

It is difficult to compare differences between mutants in phenotypic severity at a glance. I would suggest that the authors use a stacked vertical bar chart (100% cumulative bar chart).

This was an excellent suggestion—we have added the requested charts of each Figure.

3. Fig. 6I

Using AlphaFold and AlphaFold-Multimer, it is predicted that the conserved region of Afadin (1403-1455) forms an alpha-helix and interacts with alpha-E-catenin M3. Since alpha-E-catenin is an important tension transducer of adherens junctions, it would be useful to discuss whether its interaction with Afadin affects its conformation, dimerization state, and binding to other alpha-E-catenin-binding partners (such as beta-catenin, vinculin, and F-actin).

This is a good point—we have added mention of this to the revised Discussion, as follows: “Since alpha-catenin is a critical player in sensing and transducing tension at AJs [2], it will be important in the future to test whether its interaction with Afadin affects its conformation, dimerization state, and binding to other alpha-catenin-binding partners, such as beta-catenin, vinculin, and F-actin. Mutational analysis of this motif in cultured mammalian cells would be one route forward.”

Prompted by this suggestion we also have added further discussion of the potential for interaction between Drosophila Canoe and alpha-catenin later in that section of the Discussion, as follows: “In fact, there are no long conserved motifs N-terminal to the two predicted alpha-helices that comprise the potential F-actin binding site in Drosophila. However, mutations altering the M-region of Drosophila alpha-catenin alter the recruitment of Cno to tricellular junctions [55], consistent with some sort of interaction, direct or indirect. Thus, it will be important to determine if Drosophila Cno directly interacts with alpha-catenin, and to define the mechanisms by which alpha-catenin regulates Cno recruitment to AJs under tension.”

Typographical error

p. 26, l.582-583 seems to be repetitive.

"comparing one of our putative null alleles with an early stop codon, cnoR10, with one of our alleles with the alleles with the strong phenotype, cnoR17."

Fixed—thanks. It now reads: “We furthered this analysis of the zygotic phenotypes by comparing the effects on cell shapes and morphogenesis on embryos during and after dorsal closure in one of our putative null alleles with an early stop codon, cnoR10, with one of our alleles with the alleles with the strong phenotype, cnoR17.”

---

## [Editor Report · Decision Letter 1]

14 Jul 2023

Exploring the evolution and function of Canoe’s intrinsically disordered region in linking cell-cell junctions to the cytoskeleton during embryonic morphogenesis

PONE-D-23-06606R1

Dear Dr. Peifer
Mark,

We’re pleased to inform you that your manuscript has been judged scientifically suitable for publication and will be formally accepted for publication once it meets all outstanding technical requirements.

Kind regards,

Giovanni Messina

Academic Editor

PLOS ONE
---

## [Editor Report · Acceptance letter]

26 Jul 2023

PONE-D-23-06606R1 

Exploring the evolution and function of Canoe’s intrinsically disordered region in linking cell-cell junctions to the cytoskeleton during embryonic morphogenesis 

Dear Dr. Peifer:

I'm pleased to inform you that your manuscript has been deemed suitable for publication in PLOS ONE. Congratulations! Your manuscript is now with our production department. 

Kind regards, 

on behalf of

Dr. Giovanni Messina 

Academic Editor

PLOS ONE